# Actomyosin meshwork mechanosensing enables tissue shape to orient cell force

Soline Chanet[1], Callie J. Miller[2], Eeshit Dhaval Vaishnav[1], Bard Ermentrout[3], Lance A. Davidson[2,4,5] & Adam C. Martin[1]

Sculpting organism shape requires that cells produce forces with proper directionality. Thus, it is critical to understand how cells orient the cytoskeleton to produce forces that deform tissues. During *Drosophila* gastrulation, actomyosin contraction in ventral cells generates a long, narrow epithelial furrow, termed the ventral furrow, in which actomyosin fibres and tension are directed along the length of the furrow. Using a combination of genetic and mechanical perturbations that alter tissue shape, we demonstrate that geometrical and mechanical constraints act as cues to orient the cytoskeleton and tension during ventral furrow formation. We developed an *in silico* model of two-dimensional actomyosin meshwork contraction, demonstrating that actomyosin meshworks exhibit an inherent force orienting mechanism in response to mechanical constraints. Together, our *in vivo* and *in silico* data provide a framework for understanding how cells orient force generation, establishing a role for geometrical and mechanical patterning of force production in tissues.

[1] Department of Biology, Massachusetts Institute of Technology, Cambridge, Massachusetts 02142, USA. [2] Department of Bioengineering, University of Pittsburgh, Pittsburgh, Pennsylvania 15260, USA. [3] Department of Mathematics, University of Pittsburgh, Pittsburgh, Pennsylvania 15260, USA. [4] Department of Developmental Biology, University of Pittsburgh, Pittsburgh, Pennsylvania 15260, USA. [5] Department of Computational and Systems Biology, University of Pittsburgh, Pittsburgh, Pennsylvania 15260, USA. Correspondence and requests for materials should be addressed to A.C.M. (email: acmartin@mit.edu).

Forces that shape cells and tissues can be produced by the contraction of actin filament (F-actin) meshworks by the molecular motor Myosin II (myosin). The magnitude, direction and timing of contractile force depend on the organization of the cellular actomyosin meshworks and how these networks are connected between cells at the level of the tissue. How cells determine the direction in which they apply force is of critical importance to cell and tissue shape changes. Studies using individual cells on micropatterned substrates have revealed that the cytoskeleton responds to applied forces, patterns of adhesion, cell geometry and stress that feedback on the directionality and magnitude of forces generated by the network[1–5]. Whether this responsiveness to external force and geometry is important to orient force generation during the development of a tissue is poorly understood.

Previous work has shown that gradients or asymmetries in biochemical signals at the cell and tissue level can serve as instructive cues to pattern force generation. For example, in *Drosophila*, pair-rule genes expressed in stripes along the anterior–posterior (a-p) axis of the embryo regulate the expression of the Toll family receptors and the planar-polarized enrichment of Myosin II (refs 6–8). In the *Drosophila* thorax, opposing gradients of Dachsous and Four-jointed expression, which are constituents of the Fat/Dachsous/Four-jointed planar cell polarity pathway, result in the polarized localization of the myosin Dachs[9]. These systems demonstrate how molecular signals can polarize the actomyosin cytoskeleton and its upstream regulators, directing force generation across a tissue.

In addition to biochemical signals, cells embedded in tissues respond to mechanical constraints[10–13]. For example, purely physical mechanisms can account for branching morphogenesis of the embryonic mouse airway epithelium, when cultured *ex vivo* in the absence of biochemical patterning[14]. Similarly, the looping of the gut and the pattern of villi are determined by physical buckling[15,16]. These examples illustrated the passive response of tissues to external constraints, such as those imposed by differential growth. How epithelial cells actively respond to mechanical constraints by adjusting how they generate force is important to understand tissue morphogenesis.

In the *Drosophila* embryo, apical constriction in a strip of epithelial cells along the ventral midline results in the folding of the tissue and the internalization of ventral cells, forming a ventral furrow (VF). The internalized ventral cells give rise to the mesoderm that differentiates into muscle and other internal cell types such as the fat body, macrophages or lymph glands. During VF formation, epithelial tension is predominantly directed along the axis of the furrow (a-p axis) and the tissue contracts and folds orthogonal to the furrow (along dorsal–ventral (d-v) axis) (Fig. 1a)[17]. This polarized tension is associated with supracellular actomyosin fibres oriented along the direction of tension. Interestingly, the G protein-coupled receptor signalling pathway that is induced in ventral cells and activates actomyosin contraction, promoting a long, narrow VF, is also induced in posterior cells where actomyosin contraction induces a cup-like invagination of the posterior endoderm, called the posterior midgut (PMG, Fig. 6a)[18–20]. How the same signalling pathway and the same contractile machinery induce different forms in these distinct tissues is unknown. Here, we show that mechanical constraints imposed by embryo shape acts as a cue to orient the contractile actomyosin meshwork and epithelial tension to specify proper tissue folding.

## Results

**Tissue shape orients epithelial tension.** The early *Drosophila* embryo is ellipsoid with a long a-p axis and a short d-v axis. The VF domain, defined by the expression of two transcription factors Twist and Snail, is rectangular (around 18 cells along d-v and 70 cells along a-p)[21] (Fig. 1a). During VF formation, epithelial tension is predominantly directed along the long (a-p) axis of the embryo[17], consistent with the formation of a long and narrow furrow[22,23]. The tissue exhibits more movement along the d-v direction towards the ventral midline than in the a-p direction, the length of which remains almost constant (Fig. 1a). Previous work suggested that the geometry of the contractile tissue is crucial for proper tissue folding during this process[17,24,25]. However, these studies did not examine the effect of geometry on force production. Here, we investigated whether mechanical constraints imposed by tissue shape affect cytoskeletal organization and force production by cells.

To determine how tissue shape affects epithelial tension, we altered the shape of the contractile domain in two complementary ways. First, we partially expanded the number of cells adopting the ventral fate around the embryo circumference by depleting the ventral fate inhibitor Spn27A (*Spn27A* RNA interference, *Spn27A-RNAi* )[26] (Fig. 1b). In addition, we made round embryos, in contrast to ellipsoid, by inhibiting polarized oocyte elongation by knocking down the atypical cadherin *Fat2* (*Fat2-RNAi*) in somatic cells of the female ovary (Fig. 1b)[27]. *Fat2-RNAi* embryos are shorter along the a-p axis and, because of the altered shape, both ventral and dorsal surfaces have a higher curvature than in wild-type embryos. Individual cells displayed smaller mean apical area at the onset of VF formation (Fig. 2a,b); however, the number of Snail-positive cells around the circumference is not affected ($18.3 \pm 2.2$ Snail-positive cells, 33 *Fat2-RNAi* embryos compared with $16.6 \pm 2.6$ Snail-positive cells, 9 wild-type embryos), and the a-p patterning of *Fat2-RNAi* embryos is preserved (Supplementary Fig. 1). Thus, *Spn27A-RNAi* and *Fat2-RNAi* effectively made the contracting tissue less rectangular, either by expanding the short axis or shortening the long axis. Both modifications delayed furrow formation with the furrow adopting irregular shapes ($n = 39$ *Spn27A-RNAi* embryos, $n = 35$ *Fat2-RNAi* embryos, Supplementary Movie 1). In some cases, the VF failed to invaginate along the a-p axis ($n = 12/39$, *Spn27A-RNAi* embryos and $n = 3/35$ *Fat2-RNAi* embryos failed to form a furrow, Supplementary Movie 2). In *Spn27A-RNAi* embryos, we sometimes observed furrows form and invaginate perpendicular to the a-p axis (Supplementary Movie 2). Interestingly, the majority of *Fat2-RNAi* embryos hatched ($n = 88/100$ *Fat2-RNAi* embryos compared with $n = 97/100$ wild-type embryos), giving rise to larvae that were phenotypically similar, but smaller than wild-type larvae. This suggests that the round shape of the embryo did not prevent further embryogenesis. *Spn27A* loss of function, however, is lethal[26].

To test how these perturbations affect epithelial tension, we inferred relative tensile forces in three different ways and found that both perturbations resulted in more isotropic epithelial tension. First, we measured the initial recoil velocity after laser ablation[28] along the two different axes of the embryo. Initial recoil velocity is equal to pre-existing tensile forces present in the tissue before cutting divided by the viscous drag. We made 30 μm long laser incisions along either the d-v or a-p axes and followed the displacement of apical myosin, which is an established method for inferring cytoskeletal tension[29]. We calculated the initial recoil velocity (v0) of the cut edges along a-p and d-v, respectively (Fig. 1c,d). The 30 μm incisions spanned multiple cells, so our measurements would not be influenced by specific structures in individual cells. In control embryos (*Control-RNAi* or *Ctr-RNAi* embryos), the recoil velocity is approximately twofold greater along a-p than along d-v[17]. This difference is lost in *Spn27A-RNAi* or *Fat2-RNAi* embryos, indicating that these embryos generate isotropic tension during VF formation

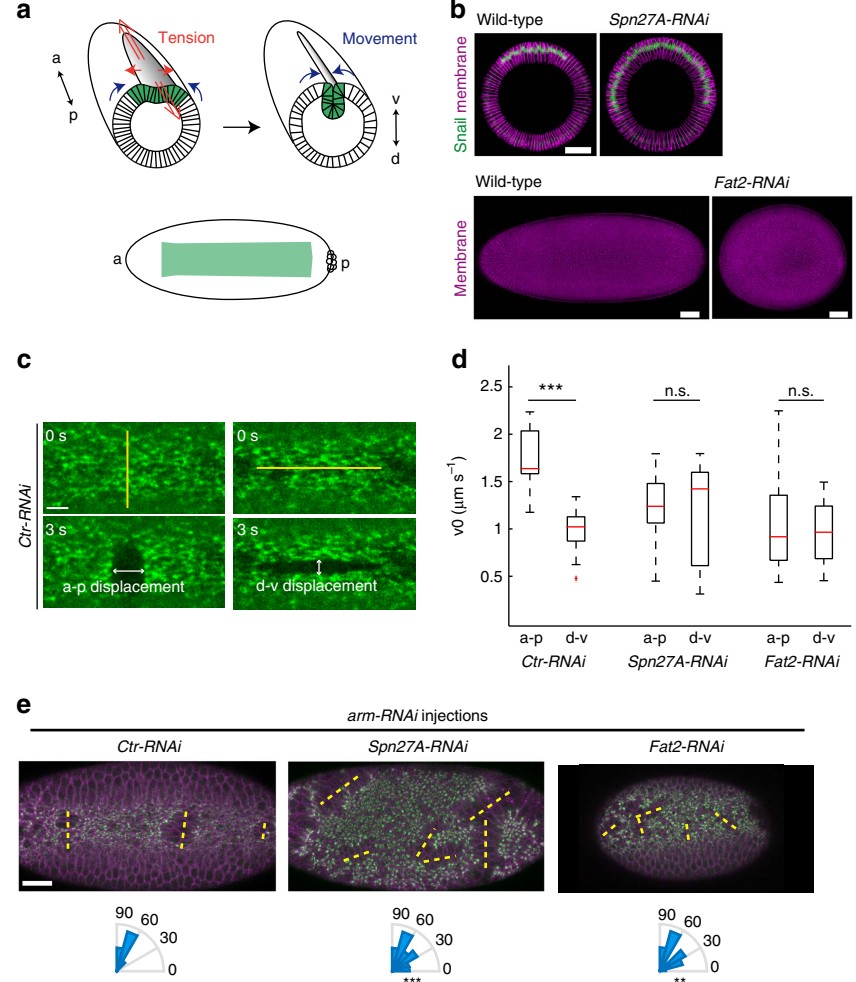

**Figure 1 | Tissue shape defines the direction of tension.** (**a**, top) Schematic representation of a gastrulating *Drosophila* embryo; the prospective mesoderm is depicted in green. Anisotropic epithelial tension (red arrows) accompanies ventral furrow formation. Tension is predominantly directed along the a-p axis, whereas there is more movement or tissue flow (blue arrows) in the d-v direction towards the ventral midline. (**a**, bottom) The VF domain (green) consists of a ventral rectangle defined by the expression pattern of Twist and Snail. (**b**, top) Depletion of *Spn27A* activity in the early embryo increases the number of cells adopting the ventral fate, marked by Snail expression (green), around the embryo circumference. (**b**, bottom) Fat2 depletion in the female ovary inhibits oocyte elongation and leads to the formation of round eggs. (**c**) Initial recoil velocity was measured along a-p after vertical line incision (yellow line) and along d-v after horizontal line incision. Images are a representative example from live embryos before (0 s) and 3 s after line ablation performed in *Ctr-RNAi* embryos expressing sqh::GFP (myosin, green). (**d**) Box-and-whisker plot of initial recoil velocity (v0) along a-p and d-v, measured after line ablation performed in *Ctr-RNAi* (n = 15 vertical incisions and n = 9 horizontal incisions), *Spn27A-RNAi* (n = 24 vertical incisions and n = 9 horizontal incisions) and *Fat2-RNAi* (n = 19 vertical incisions and n = 11 horizontal incisions) embryos. ***P < 0.001, n.s., not significant, t-test. (**e**) Images from live embryos injected with *arm* double-stranded RNA (*arm-RNAi*) to lower junctional complexes (top panels) in *Ctr-RNAi*, *Spn27A-RNAi* and *Fat2-RNAi* embryos. The *arm-RNAi* resulted in tissue tearing and separation of the myosin meshwork (green). Yellow lines highlight the direction of the tears. The angles between the tears and the a-p axis were quantified (lower panels). Angle repartitions are significantly different in *Spn27A-RNAi* (n = 26 tears, 3 embryos) and *Fat2-RNAi* (n = 22 tears, 3 embryos) compared with *Ctr-RNAi* embryos (n = 13 tears, 3 embryos). **P < 0.01, ***P < 0.001, t-test. Scale bars, 40 μm (**a,e**) and 10 μm (**c**). All images of *en face* views show the ventral side of the embryos, and anterior is left and posterior is right.

(Fig. 1d). To confirm this result, we also induced isotropic (circular) wounds in live embryos and measured the aspect ratio of the wound after stretching that has been used previously to examine stress anisotropy in tissues[30]. In control embryos, wounds became elliptical with their long axis oriented along the a-p axis; however, wounds opened symmetrically in *Spn27A-RNAi* and *Fat2-RNAi* embryos (Supplementary Fig. 2 and Supplementary Movie 3). In an independent approach to probe tension directionality, we analysed the directionality of tissue tears after sporadic disruption of intercellular contacts. Reducing the level of adherens junction (AJ) components weakens the mechanical integrity of the epithelium, making it prone to fail under tension. Actomyosin contraction induces epithelial tears

perpendicular to the direction of highest tension in AJ-depleted embryos[17]. We injected RNAi against *armadillo* (*arm*, the fly β-catenin) that reduced AJ levels and locally induced tissue tears. Consistent with the laser ablation results, there was no longer a preferential orientation for tissue-wide tearing events upon reducing AJ levels in *Spn27A-RNAi* and *Fat2-RNAi* (Fig. 1e and Supplementary Movie 4). Note that the higher number of tears likely reflects the slower invagination in *Fat2-RNAi* embryos and the greater area of contractile tissue in *Spn27A-RNAi* embryos. Thus, altering the geometry of the contractile domain by modifying gene expression pattern or embryo shape changes the directionality of tensile forces during VF formation.

**Tissue shape is critical for proper cell shape changes.** We next asked how isotropic tension in the VF altered individual cell behaviour. We previously showed that forces transmitted between cells of the VF tissue promote anisotropic cell constriction, with constriction predominantly occurring along the d-v axis, resulting in asymmetric shape[17]. When tissue shape is modified in *Spn27A-RNAi* and *Fat2-RNAi*, cell contraction appeared to stall at a larger apical area compared with control embryos (Fig. 2a). In addition, apical constriction was more isotropic in both cases, suggesting that external forces resisting constriction were equalized along both axes, consistent with isotropic tension (Fig. 2b,c). To determine whether the differences in cell shape changes we observed in *Fat2-RNAi* and *Spn27A-RNAi* embryos were indeed due to changes in the mechanical environment of the cells and not to altered intrinsic force generation, we tested whether normal cell constriction occurred when groups of cells were mechanically disconnected from the surrounding tissue. We first checked that AJ proteins, which mechanically link cells, localized normally in *Spn27A-RNAi* and *Fat2-RNAi* (Supplementary Fig. 3). Then, we performed *arm-RNAi* and we followed constriction of groups of cells located between epithelial tears (Fig. 2d). Disrupting tissue integrity, and thus external constraints to constriction imposed by tissue shape, restored full cell constriction and normal constriction rate in *Spn27A-RNAi* and *Fat2-RNAi* (Fig. 2e). It also induced rapid isotropic cell constriction in both wild-type as previously shown[17] and RNA interference (RNAi) conditions. This indicated that our RNAi conditions do not affect the cells' intrinsic ability to constrict, but rather the external resistance to constriction of cells within the contractile domain. Therefore, altering tissue shape alters the mechanical constraints to individual cell constriction within the VF domain that affects how cells constrict.

**Geometric constraints regulate actomyosin organization.** External forces and cell geometry have been shown to influence actomyosin organization in individual cells spread on an extracellular matrix substrate that feedback on the directionality of force generation by the network[2,31,32]. To determine whether such a mechanism could contribute to changes in epithelial tension we observed in an embryo, we examined the actomyosin cytoskeleton, which is the molecular machine that generates tension. In *Ctr-RNAi* embryos, we observed the formation of an actomyosin meshwork that was highly interconnected from cell to cell. This tissue-wide meshwork often exhibits supracellular actomyosin fibres that are oriented along the a-p axis. Importantly, these oriented actomyosin fibres appeared in cells before the cells themselves were fully constricted, appearing when cells only became slightly anisotropic, suggesting that this cytoskeletal organization precedes the full constriction of the cell (Supplementary Movie 5). In contrast, in *Spn27A-RNAi* or *Fat2-RNAi* embryos, oriented actomyosin fibres did not form; instead myosin became organized into prominent rings centred within the apical domain of cells (Fig. 3a, yellow arrows). The continuity of the supracellular meshwork is maintained by small radially oriented fibres that connected rings from cell to cell (Fig. 3a, red arrowheads). Similarly, F-actin organization was depleted from the apical centre, such that it resembled a ring with actin enriched around the apical periphery and a hole in the middle of the apical cortex (Supplementary Fig. 4a,b).

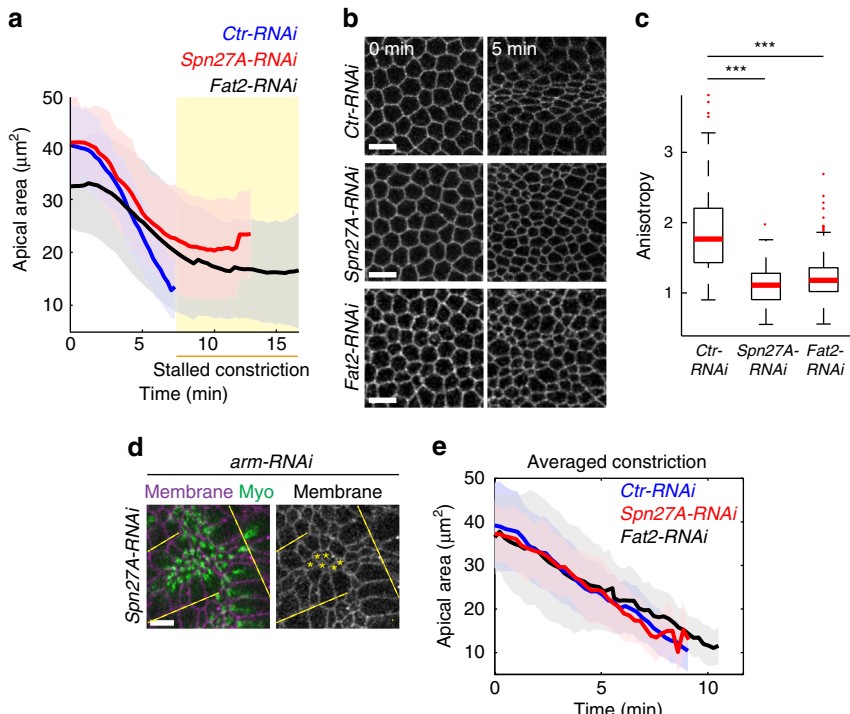

**Figure 2 | Tissue shape is critical for proper cell shape changes.** (**a**) Mean apical cell area over time calculated for *Ctr-RNAi* (*n* = 119 cells, 2 embryos), *Spn27A-RNAi* (*n* = 134 cells, 2 embryos) and *Fat2-RNAi* (*n* = 418 cells, 3 embryos) embryos during VF formation. Shaded area indicates s.d. (**b**) Images of VF cell outlines in live embryos depleted for the indicated gene before and 5 min after the onset of cell constrictions. (**c**) Box-and-whisker plots of cell anisotropy 7 min after the onset of cell constriction for the embryos analysed in **a**. Cell anisotropy is the length of the cell along the horizontal axis divided by the vertical length, such that cells elongated along the a-p axis exhibit anisotropy > 1. Isotropic is anisotropy = 1. (**d**) Injection of *arm-RNAi* locally induced tissue tears (dotted lines) that disrupted tissue integrity. Cell constriction was followed for cells between tears (asterisks). (**e**) Mean apical area over time for 16 *Ctr-RNAi*, 18 *Spn27A-RNAi* and 15 *Fat2-RNAi* constricting cells between tears after *arm-RNAi* injection. Shaded area indicates s.d. Scale bars, 10 μm. ***P < 0.001, *t*-test.

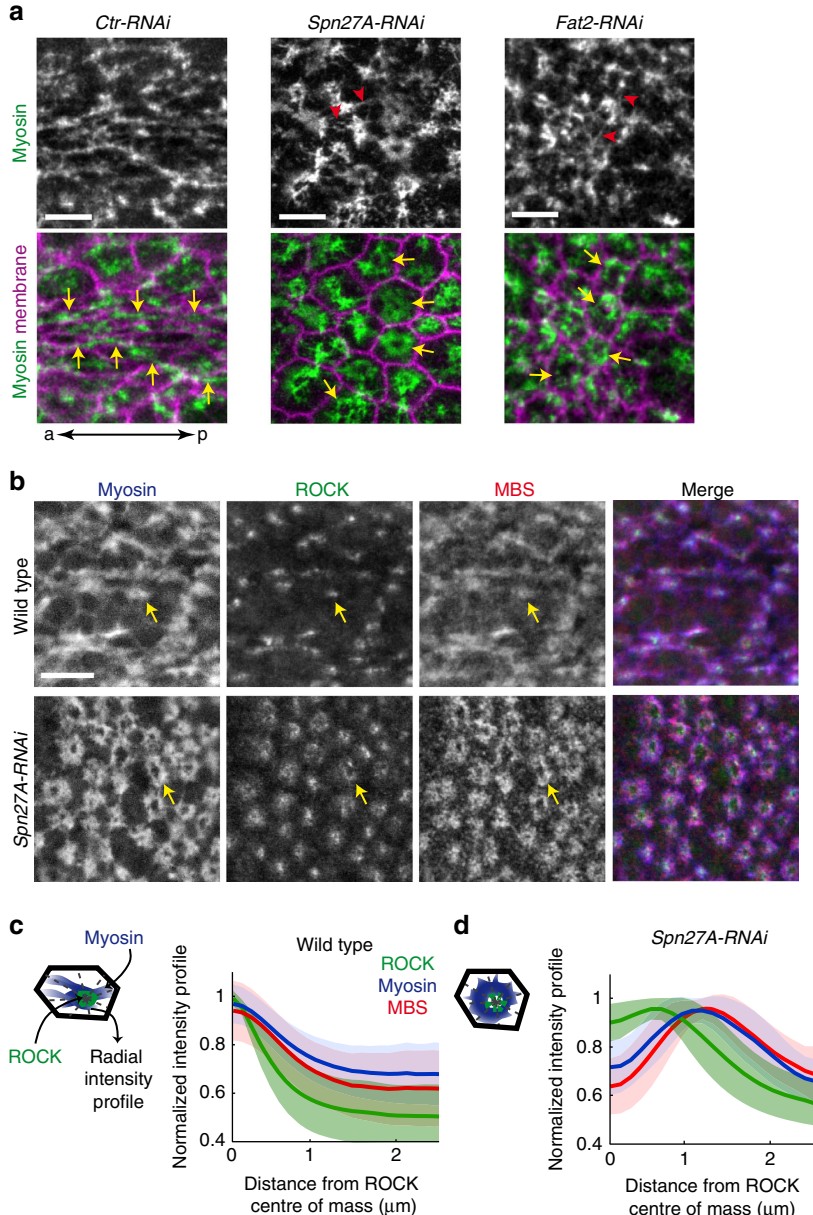

**Figure 3 | Change in tissue shape induces alternative configuration of the ROCK-myosin signalling module.** (**a**) Apical myosin meshwork (green) organized into a-p oriented fibres during VF formation in *Ctr-RNAi* embryos (arrows) but organized into rings in *Spn27A-RNAi* and *Fat2-RNAi* embryos (arrows). Red arrowheads point to radially oriented spokes that connected myosin rings from cell to cell. a, anterior; p, posterior. (**b**) Images of VF cells in fixed wild-type and *Spn27A-RNAi* embryos showing the localization of ROCK::GFP (green) sqh::mCherry (blue) and MBS (red). Myosin and MBS formed a fibrous network and ROCK a condensed focus in wild-type cells (arrow), whereas Myosin, MBS and ROCK organized into rings in *Spn27A-RNAi* cells (arrow). (**c**,**d**) Quantification of ROCK, myosin and MBS polarization in the apical domain of VF cells as the signal intensity profile from the centre of ROCK density to the apical margin. Left, schematic of cells with radial lines emanating from the ROCK centre of mass. Right, mean normalized intensity profile of ROCK, myosin and MBS in wild-type cells (**c**, n = 152 cells, one embryo) and *Spn27A-RNAi* cells with clear ROCK rings (**d**, n = 27 cells, one embryo). Shaded area indicates s.d. Scale bars, 5 μm (**a**) and 10 μm (**b**).

Upstream regulators of myosin also became reorganized into a ring-like configuration. Myosin is regulated by the opposing actions of the Rho-associated kinase (ROCK) and the myosin phosphatase that phosphorylate/activate and dephosphorylate/inactivate the myosin II regulatory light chain, respectively[33,34]. In wild-type cells, ROCK is concentrated in the centre of the apical domain and colocalized with myosin fibres (Fig. 3b)[35,36]. In contrast, associated with myosin ring formation in *Spn27A-RNAi* embryos, ROCK often localized into rings that were concentric to myosin (Fig. 3b). To quantify the localization of ROCK relative to myosin, we measured the radial intensity profile of different

proteins relative to the centre of mass of ROCK signal intensity. In wild-type cells, myosin, ROCK and myosin phosphatase (stained with MBS, myosin-binding subunit of the myosin phosphatase) all show a peak at the centre of mass of ROCK intensity (Fig. 3c). In *Spn27A-RNAi* VF cells, we distinguished three types of ROCK organization, condensed foci, rings or diffuse foci (Supplementary Fig. 4c,d). In cases where ROCK rings were observed, ROCK localized to the inside edge of a broader myosin and myosin phosphatase ring (Fig. 3b,d). Because ROCK localization depends on its own activity and the actin cytoskeleton[37–39], the formation of ROCK rings/foci could

follow the actomyosin reorganization. Thus, it appeared that changes in mechanical pattern imposed by tissue shape cause reorganization of myosin, actin, ROCK and myosin phosphatase into rings.

We did not observe differences in the early behaviour of myosin during the growth of the meshwork between *Ctr-RNAi*, *Spn27A-RNAi* and *Fat2-RNAi* embryos. Initiation of myosin assembly and total mean apical myosin levels were similar between conditions (Supplementary Fig. 5a). Like wild-type embryos, myosin assembly occurred in pulses, and we measured similar pulsatile dynamics in *Ctr-RNAi*, *Spn27A-RNAi* and *Fat2-RNAi* embryos (Supplementary Fig. 5b–e). Thus, our data are consistent with the initial intrinsic contractile properties of cells not being modified in our RNAi conditions, but rather the actomyosin meshworks adopting different conformations in response to altered mechanical constraints to contraction.

**Actomyosin meshworks respond to mechanical constraints**. If actomyosin meshworks respond to mechanical constraints imposed by tissue shape, then the formation of actomyosin rings in *Spn27A-RNAi* should be suppressed by artificially decreasing resistance along the d-v axis, thus mimicking the asymmetric mechanical pattern in wild-type embryos. Repetitive laser cutting along two horizontal lines severed apical actomyosin meshworks (Fig. 4a). This micropatterned laser cutting procedure therefore mechanically disconnected cells between the two cuts from the dorsal part of the embryo, releasing resistance to constriction along the d-v axis without affecting resistance to constriction along the a-p axis. As a read-out for the mechanical pattern that resisted cell constriction, we measured cell anisotropy over time; cells at the midline constricted preferentially along the d-v axis and formed wedge shapes characteristic of wild-type cells, suggesting that the laser incisions decreased resistance along the d-v axis in *Spn27A-RNAi* embryos (Fig. 4b,c and Supplementary Movie 6). This mechanical perturbation in *Spn27A-RNAi* embryos prevented the formation of actomyosin rings in the central cells. We observed instead that myosin became organized into supracellular myosin fibres oriented along the a-p axis even before the cells fully constrict, similar to wild-type embryos ($n = 16$ embryos, Fig. 4a,d and Supplementary Movies 6 and 7). Importantly, myosin still formed rings in regions away from the cuts where cells were not expected to be freed from the dorsal tissue by the incisions (Fig. 4e).

Furthermore, lowering anisotropic tension along the a-p axis by repeatedly making two vertical incisions in wild-type embryos with ellipsoid shape was associated with isotropic constriction and the formation of myosin rings (Fig. 4f–i and Supplementary Movies 8 and 9; $n = 20$ embryos). These mechanical manipulations strongly support the model that actomyosin meshworks adapt to external mechanical pattern and redirect cell forces.

**Constraints inherently orient network forces *in silico***. Previous work demonstrated how boundary stiffness can influence the collective behaviour of filament/motor systems to influence force production by these networks[40–42]. However, these models did not investigate the effects of anisotropic boundary conditions on an apical-like two-dimensional contractile cortex. To gain insight into how apical actomyosin arrays respond to external forces, we developed an *in silico* model of the actomyosin cortex with modifiable boundary constraints. In this model, F-actin was represented as polar filaments that turnover at a given rate, consistent with the presence of F-actin turnover in VF cells[43]. Myosin motors were represented as bipolar filaments, each end of which could bind and move directionally (towards plus end) along actin filaments (Fig. 5a); free parameters, whose values were

based on available published data describing the rates of attachment and detachment from actin filaments (see Supplementary Methods, Supplementary Table 1). Myosins exerted force on actin filaments when the heads walked away from each other, stretching a spring connecting the heads that resulted in restoring forces that translated and/or rotated the actin filaments, contracting the network (Fig. 5a). To model symmetric or asymmetric mechanical constraints to network contraction, we embedded our simulation within a defined space, where actin filaments whose plus ends were within one-half of a filament length from the boundary were coupled to the boundary via a spring-like attachment with a defined stiffness representing the compliance or tension of surrounding tissues (Fig. 5b). The orientation of the 'boundary filaments' was consistent with the *in vivo* observation that actin filament plus ends are enriched at cell junctions[39]. These 'boundary filaments' were free to translate, rotate and turnover, like the actin filaments not attached to the boundary (called free filaments) and were disconnected if the plus end moved more than one-half filament length from the boundary. The stiffness of the boundary springs could be modulated to impose different resistances and were also used to calculate force generated at specific regions of the boundary.

Our model predicted that altering the stiffness of the boundary affects the force produced by the simulated network. Absence of a boundary constraint resulted in the rapid contraction of motors into an aster-like structure in the middle of the cell (Fig. 5d) that resembled *in vivo* observations of myosin flow in cells that lack strong mechanical adhesion to their neighbours[17]. Introduction of stiff springs (representing a 100 kPa elastic continuum) around the entire boundary slowed this inward movement (Fig. 5d), and resulted in the formation of a ring of actin and myosin that generated isotropic force around the boundary (Fig. 5c,e). Interestingly, when boundary springs along one axis of the space were made softer than the orthogonal axis (for example, softer boundary filaments of 20 kPa along the top/bottom faces modelling anisotropic resistance to contraction) we observed the meshwork flatten along the soft axis, adopting an oval structure aligning with the long axis and the network directed force along the axis with stiffer boundary springs (Fig. 5c, left panels and Fig. 5e). Note that in all cases, filaments were still present and exerted force on all boundaries even those bearing softer springs (for example, the case of anisotropic boundary). We measured the emergence of force directionality by measuring the mean ratio of forces produced along the stiff axis (left and right side of the boundary) over forces produced along the soft axis (top and bottom faces of the boundary) as a function of the resistance asymmetry of the boundary (that is, increasing ratio in boundary spring stiffness along left/right versus top/bottom faces). Strikingly, force directionality emerged when anisotropy in boundary resistance exceeded somewhere between a 1.7- and 5-fold difference along two axes (Fig. 5c) and always aligned along the axis of greater resistance to constriction (that is, the axis with stiffer boundary springs). Force directionality emerged from a decrease in the amount of force produced along the axis of least resistance (Supplementary Fig. 6a,b). Thus, our *in silico* model of actomyosin contraction predicted that actomyosin networks inherently adjust the direction of force in response to asymmetric constraints to contraction.

To gain insight into what properties of the meshwork contribute to orient force production, we examined motor alignment. At each time point, motors connecting two filaments have an angle (between 0° and 180°) corresponding to the orientation of the motor with respect to the horizontal axis in the simulation (Fig. 5f). To test whether constraints to contraction influence motor angle, we compared the number of force-generating ($\geq 0.3$ nN) motors aligned along the stiff axis (between

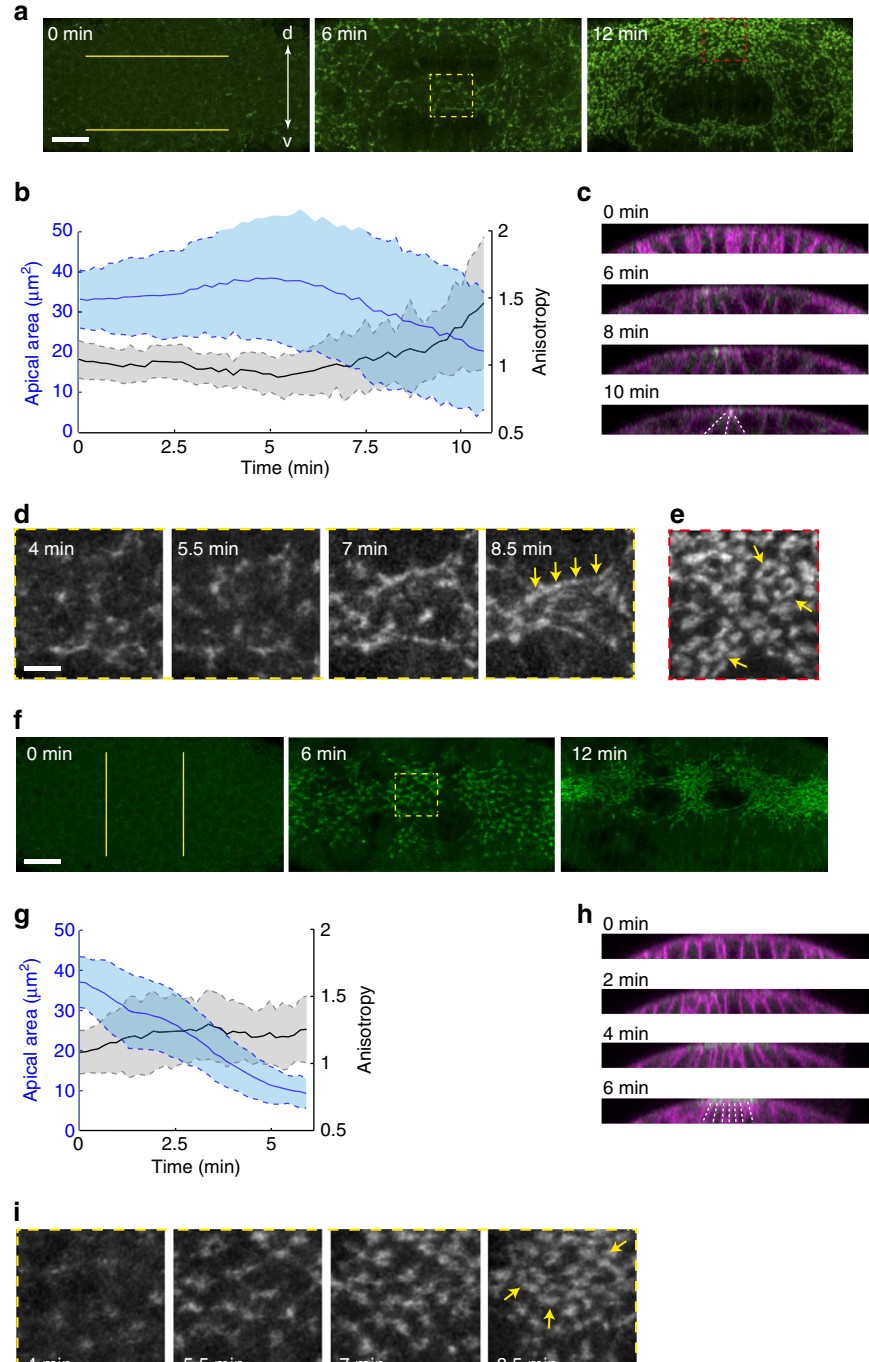

**Figure 4 | Actomyosin meshworks respond to mechanical constraints.** (**a**) Images from live *Spn27A-RNAi* embryos showing myosin organization (green), while two horizontal incisions of 60 μm long (yellow lines) were performed repetitively using a 2-photon laser to lower resistance along the d-v axis. (**b**) Mean apical cell area (blue) and anisotropy (black) over time for 45 cells located in between two horizontal incisions in a *Spn27-RNAi* embryo. Shaded area indicates s.d. (**c**) YZ cross-section view of the same embryo used for quantification in **b**. The wedge-shape of the two medial-most invaginating cells is highlighted with white dotted lines. (**d**) Time-lapse images showing myosin fibre formation over time in the region within the yellow dotted box depicted in **a**, middle panel. (**e**) Higher magnification of region within the red dotted box in **a**, right panel, showing myosin rings (arrows) in a region outside of the horizontal cuts. (**f**) Images from live *Ctr-RNAi* embryos showing myosin organization (green), while two vertical incisions of 50 μm long (yellow lines) were performed repetitively using a 2-photon laser to reduce resistance to contract along the a-p axis. (**g**) Mean apical cell area (blue) and anisotropy (black) over time for 65 medial cells located between two vertical incisions in a *Ctr-RNAi* embryo. Shaded area indicates s.d. (**h**) YZ cross-section view of the same embryo as in **g**. The final shape of the constricting cells is highlighted with white dotted lines. (**i**) Time-lapse images showing myosin ring formation over time in the region within the yellow dotted box depicted in **f**, middle panel. Scale bars, 20 μm (**a,f**) and 5 μm (**d,i**).

0° and 30° and 150° and 180°) to motors aligned along the soft axis (between 60° and 120°). We found a 20% enrichment of motors aligned along the stiff axis, whereas motors did not display preferred orientation when boundary stiffness was isotropic (Fig. 5f). Thus, an asymmetric mechanical constraint to contraction caused force-generating myosin motors to become

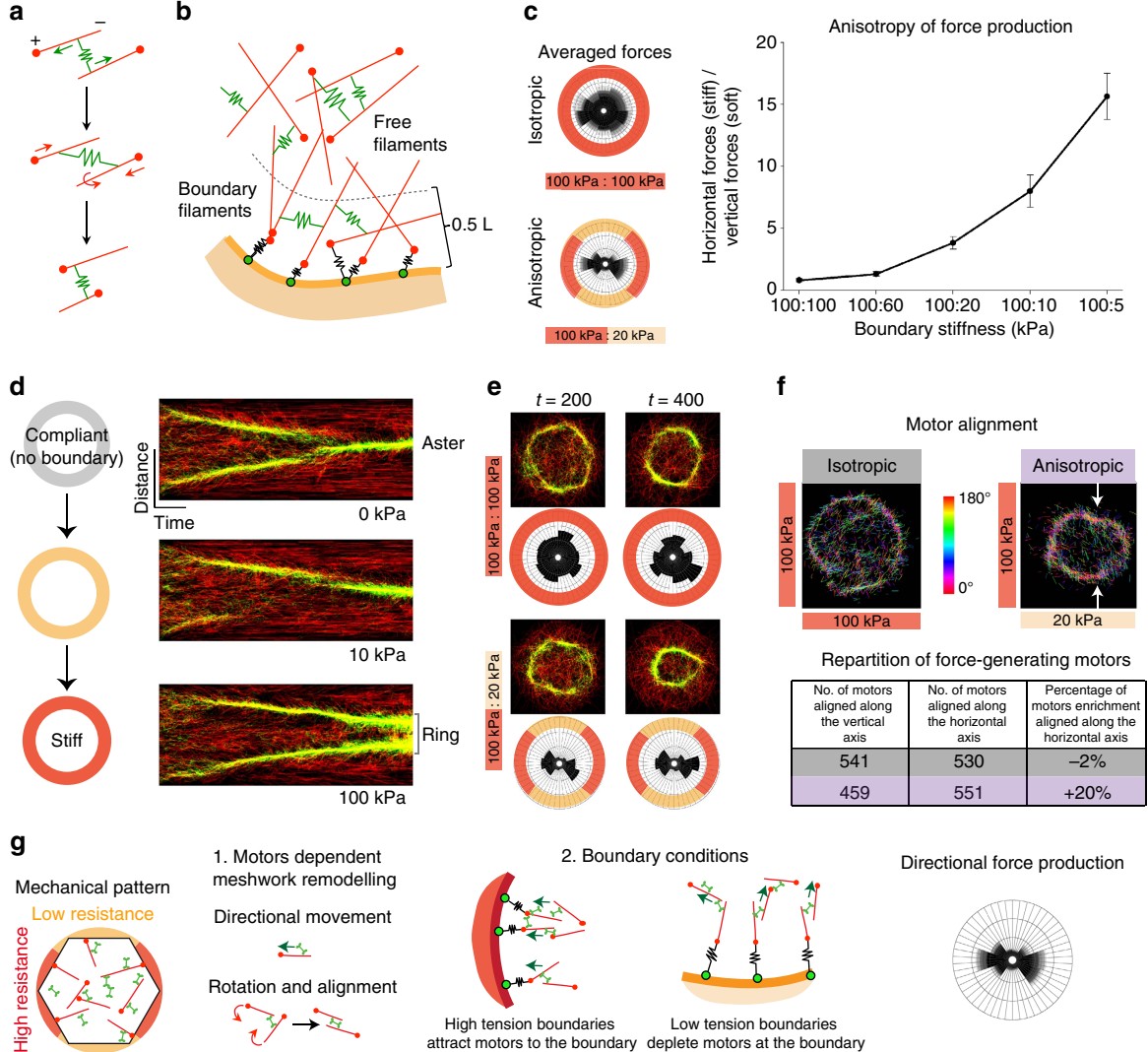

**Figure 5 | 2D *in silico* actomyosin cortex predicts that external constraints alter cytoskeleton organization and force direction. (a)** F-actin is modelled by polar filaments (red) and motors by springs (green). As motor exerts force on filaments, filaments translocate and/or rotate. **(b)** Within a perimeter of 0.5 filament length, the plus ends of the filaments are caught by springs with defined stiffness to model boundary resistance. **(c**, left) To simulate isotropic constraints, boundary springs (stiffness = 100 kPa) were equally distributed around the boundary. To simulate anisotropic constraints, softer springs (20 kPa) were placed at the top and bottom of the boundary. The rose diagrams illustrate the average force generated on boundary springs for each condition ($n \geq 10$ simulations), and circle divisions represent 10 nN. (Right) The ratio of force production along stiff (left/right) over soft (top/bottom) axes was calculated and plotted depending on increasing values of stiffness anisotropy between left/right and top/bottom boundary springs; $n \geq 10$ simulations per conditions. Error bars represent s.d. **(d)** Kymographs representing the contraction of the network into an aster (top) or the stabilization of a ring-like structure (bottom) depending on the resistance of the boundary. **(e)** Images of representative simulations at two time points showing motor (green) and actin filament (red) organization. The associated rose diagram illustrates forces generated on the boundary by these networks, and circle divisions represent 10 nN. Top, isotropic boundary. Bottom, anisotropic boundary. **(f)** Motor alignment for simulations with isotropic or anisotropic boundaries ($t = 250$ time-steps). The motors are colour coded based on angle relative to the horizontal axis, with red motors being most aligned with that axis. Anisotropic boundary constraints resulted in motors aligned with the stiff (horizontal) axis along the top and bottom of the oval-like myosin structure (arrows). Quantification of motor alignment is given in the table. Numbers highlighted in grey correspond to the isotropic case and numbers highlighted in purple to the anisotropic case. There is a 20% enrichment bias in motors aligned along the stiff horizontal axis when the resistance of the boundary is anisotropic. **(g)** Model illustrating the factors that influence motor movement and alignment, and force production in response to asymmetric boundary stiffness.

aligned along the axis of greater resistance to contraction that orients force production along this axis (Fig. 5f). In the model, this was due to the fact that it was easier for motors and filaments to move and rotate in the 'softer' axis so that they aligned with the stiff axis (Fig. 5g).

In addition, meshwork organization and activity may also depend on cell shape. Several studies have shown that cytoskeletal organization depends on the geometry of the cell and the arrangement of anchor points at the cell periphery[2,3,44]. During

VF formation, cell shapes become anisotropic, potentially influencing the organization of actomyosin meshworks along the long axis of the cell. To test the possible contribution of cell shape to force generation, we embedded our actomyosin simulation in an ellipse with stiff (100 kPa) boundary springs, mimicking anisotropic cell shape (Supplementary Fig. 6c). Interestingly, the average force produced within an ellipse was perpendicular to the long axis, in contrast to the *in vivo* orientation of force production parallel to the long axis of cells.

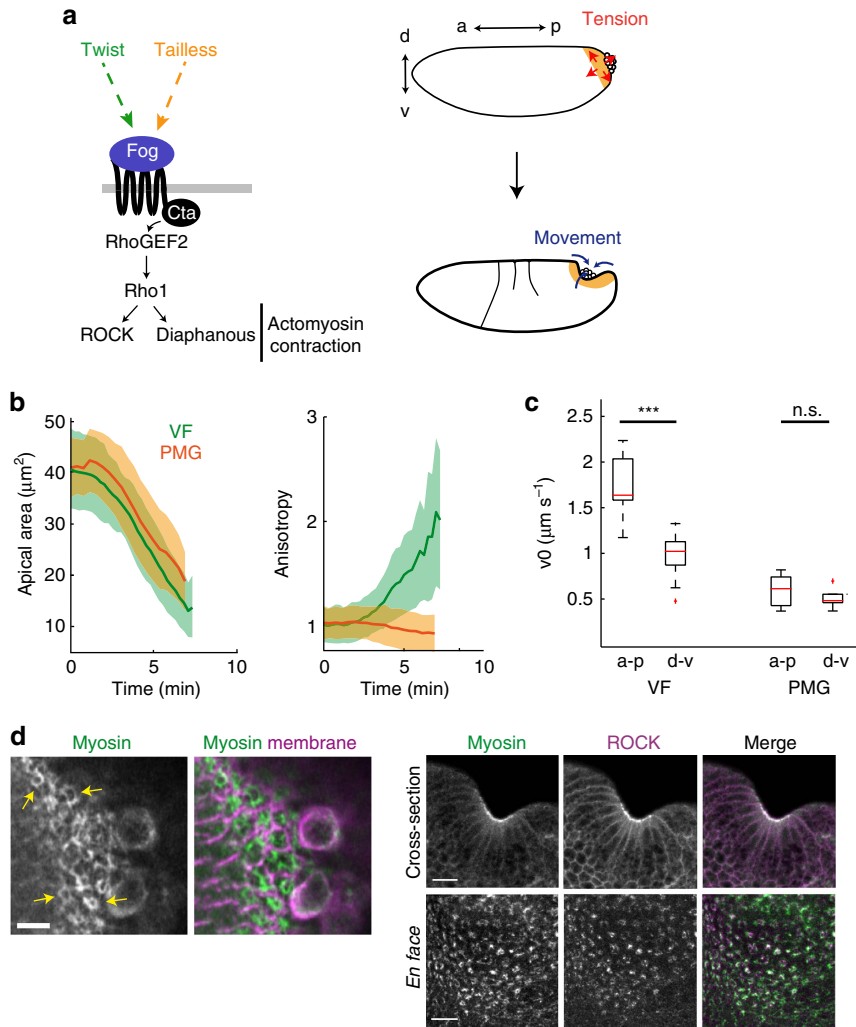

**Figure 6 | Cup-like invagination of the posterior midgut is associated with actomyosin rings and isotropic tension.** (**a**) Apical constriction is activated downstream of the Fog pathway in VF cells (where Fog is transcriptionally activated by Twist) and PMG cells (where Fog is transcriptionally activated by Tailless). PMG formation is driven by apical constriction of the posterior–dorsal cells forming a cup-like invagination. (**b**) Mean apical area (left) and anisotropy (right) over time during VF (green, $n = 119$ cells, 2 embryos) and PMG (orange, $n = 188$ cells, 3 embryos) formation. PMG cells constrict isotropically. Shaded area indicates s.d. (**c**) Box-and-whisker plot of initial recoil velocity (v0) along a-p and d-v, measured after line ablation performed in the VF ($n = 15$ vertical incisions and $n = 9$ horizontal incisions) and the PMG ($n = 7$ vertical incisions and $n = 7$ horizontal incisions). ***$P < 0.001$, n.s., not significant, $t$-test. (**d**) Left panels are fixed images of wild-type PMG cells, showing apical myosin organization in rings (arrows). Staining for subapical actin delineated cell boundaries. (**d**, right) Images of cross-section and *en face* views of PMG cells from a live embryo expressing ROCK-GFP (magenta) and sqh::mCherry (green). Scale bars, 10 μm (right panels) and 20 μm (left panels).

Thus, asymmetric mechanical constraints rather than confinement and anisotropic cell shape better explained the orientation of force generation observed *in vivo*.

Together, our data showed that actomyosin meshwork contraction exhibits mechanosensitive behaviour as it can sense and respond to constraints imposed by boundary conditions.

**Similar biochemical signals can induce distinct tissue forms.** Our model and experimental data implied that apical actomyosin meshworks could generate different force patterns depending on the mechanical constraints within the tissue. To test whether this responsiveness of actomyosin meshworks is utilized by the embryo to generate distinct tissue forms we looked at other tissue folding events. During *Drosophila* gastrulation, right after the VF has invaginated, the dorsal posterior end of the embryo flattens and invaginates in a process called the PMG invagination (Supplementary Movie 10). PMG invagination requires the same

G protein-coupled receptor signalling pathway that operates in the VF and similarly involves apical actomyosin meshwork contraction[19,20] (Fig. 6a). Myosin contraction exhibited pulses, qualitatively similar to myosin in the VF (Supplementary Fig. 7). However, we found that PMG cells constricted isotropically and generated isotropic epithelial tension while forming a cup-like indentation rather than a furrow (Fig. 6a–c and Supplementary Fig. 2b). Strikingly, we observed a ring-like organization of myosin and ROCK on the apical surface of constricting cells in the PMG (Fig. 6d), similar to that of VF cells in *Spn27A-RNAi* or *Fat2-RNAi* embryos. We suggest that the fly embryo uses the responsiveness of actomyosin meshworks to generate different force patterns and tissue forms (Fig. 7). Interestingly, our model predicts that when mechanical constraints are more isotropic during VF formation like in *Spn27A-RNAi* or *Fat2-RNAi* embryos, it results in more cup-like invagination (resembling PMG invagination). In *Spn27A-RNAi* and *Fat2-RNAi* embryos, we always observed a defect in VF formation, the ventral tissue

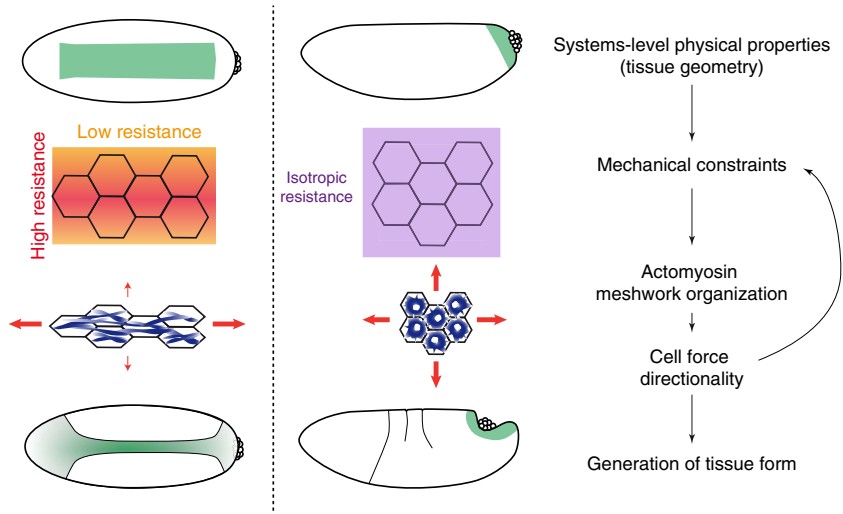

**Figure 7 | Model: tissue shape orients actomyosin meshworks and tension to generate specific tissue forms.** Tissue shape and geometry impose mechanical constraints to deformation. For example, the rectangular shape of the VF means that more cells generate tension (red arrows) and resist constriction along the long a-p axis than the short d-v axis, whereas the more isotropic domain of the presumptive PMG imposes isotropic constraints. Actomyosin meshworks sense and respond to these constraints by adopting different configurations (that is, rings or fibres). The meshwork configuration, in turn, could orient cell force generation. The combination of tissue geometry and cell force directionality then governs the final tissue form.

forming a wider and flatter indentation than wild-type tissue, sometimes resembling a cup-like indentation (for example, in Supplementary Fig. 1, the *Spn27A-RNAi* embryo stained for *sim* RNA display two cup-like indentations within its extended ventral domain).

## Discussion

Generating proper organ and organism shape requires that hundreds or thousands of cells coordinate how they generate force to elicit proper tissue form. Here, we provide evidence that tissue and even organism shape, by imposing specific mechanical constraints, can instruct cells how to direct force. We explored the role of tissue shape and mechanical pattern on cytoskeletal organization in four different ways. (1) We made the contracting VF domain less rectangular by extending the expression of Twist and Snail around the embryo circumference using *Spn27A-RNAi*. (2) We shortened the a-p axis of the embryo by generating round embryos using *Fat2-RNAi*. (3) We reduced tension along the a-p axis of wild-type embryos by performing laser incisions that separated cells from the anterior and posterior neighbours. (4) We examined constriction of a different tissue, the PMG, that relies on a similar signalling pathway, but where the contracting domain is symmetric around the posterior pole of the embryo. We showed that in each of these different cases, the cytoskeleton becomes organized into rings in the centre of the apical domain and the pattern of force is isotropic. This contrasts with wild-type VF formation, where actomyosin meshworks are organized into nodes and fibres and force is predominantly directed along the a-p axis. Importantly, actomyosin fibres and oriented forces can be restored by mechanically changing the shape of the contractile domain, and restoring the wild-type mechanical context using laser ablation. Thus, our results show that cellular force in a tissue does not always result from the genes expressed, but is also dependent on the geometrical context of the tissue. This implies that the physical properties of an organ, such as shape and geometry, can provide the information for cells to orient force and generate proper tissue form.

We envision the VF forming in the following way. The ellipsoid shape of the embryo ensures that the domain of cells that constrict has a long and short axis. More cells constricting along

the long axis introduces a mechanical constraint that resists contraction along that axis (Fig. 7). This anisotropy then influences the organization of the actomyosin meshwork in cells embedded in this tissue that affects how these cells direct force. Because actomyosin meshwork organization cannot be altered independent of tissue mechanics (and constraints) it is impossible to assess the direct contribution of the reorganization to direct force, but our *in silico* simulations predict how this might occur (see below). Cells directing tension along the axis of highest resistance would further promote anisotropic tension, reinforcing the mechanical constraint (Fig. 7). In the case of the VF, this anisotropic tension and resistance to constriction along that axis has the effect of having constriction occur mostly in the orthogonal axis. This leads to the formation of a furrow that exhibits curvature along the d-v axis.

Studies have revealed that mechanical constraints can passively affect tissue movement and shape[15,16,45]. We show here that tissue geometry does not simply impose external boundaries and constraints to deformation, but rather, cells embedded within a changing tissue actively react to and adapt how they produce force. Our data support a model whereby this adaptation relies on the ability of actomyosin meshworks to respond to mechanical constraints.

Two previously published models demonstrated a mechanosensing property of actomyosin networks that adapt and respond to substrate stiffness[40,41]. In addition, a model of contractile ring formation suggested how tension produced by myosin can move and align filaments to form a bundle[46]. Our *in silico* simulations make specific predictions as to how a two-dimensional actomyosin meshwork responds to mechanical constraints. We found that under anisotropic boundary constraints, force-generating motors align along the axis of greatest resistance and, therefore, generate greater force in this direction. The mechanosensitivity can be explained by filament rotation and translation that is facilitated along the softer axis. This prediction can be translated to the tissue level, where cells embedded in a tissue also can experience anisotropic constraints (for example, more cells constricting along the a-p axis). In the VF, the greater resistance to constriction along the a-p axis appears to lead to polarization and alignment of the actomyosin cortex. In turn, aligned motors and directed cell force would

further promote anisotropic epithelial tension. Together, our experimental and theoretical findings suggest that actomyosin meshworks can act as mechanosensors. Actomyosin meshworks are in the centre of a feedback loop, whereby they generate and respond to tissue-level pattern of epithelial tension and in this way adapt force generation to geometric and mechanical constraints (Fig. 7).

Other mechanisms can potentially reinforce this mechanical feedback loop. Myosin is also a mechanochemical enzyme and its activity and binding affinity can be influenced by mechanical force[40,47–50]. Mechanical forces could also affect the F-actin assembly and disassembly[51–53]. Finally, AJ complexes, which are at the cell's front line for receiving mechanical stimuli, are also mechanosensitive and tend to cluster and grow under load[54,55]. Although we have not noted a planar polarized enrichment of AJ complexes during VF formation, it is also possible that polarized forces stabilize the attachment between the contractile meshwork and the AJ[56]. While these other mechanosensitive steps were not included in our *in silico* model, we expect that they further contribute to the adaptive response of actomyosin meshworks.

Two-dimensional medioapical (and basal) meshworks are prevalent and have been described in a variety of different tissues and different species, and support different types of cellular processes such as apical contraction and cell intercalation[13,57–63]. We propose that these meshworks have a propensity to adapt and respond to mechanical cues by changing conformation. It is possible that these meshworks have been selected to support rapid and integrated cell and tissue shape changes, because their responsive nature provides a mechanism to coordinate and orient cell forces in a tissue. Together, our results demonstrate that mechanical and geometrical patterning can be a potent mechanism in controlling cell behaviour and sculpting tissue shape. Mechanical sensing by two-dimentional actomyosin meshworks is potentially a conserved feature of tissue development.

## Methods

**Fly stocks and genetics.** Myosin was visualized in live and fixed embryos using myosin regulatory light chain (*sqh* in *Drosophila*) fused to GFP, sqh::GFP[64] or mCherry, sqh::mCherry[57] and cell outlines were visualized using the plasma membrane marker Gap43::mCherry[17]. We used lines with fluorescent protein transgenes inserted either on the second or third chromosome. For maternal knockdown experiments, the following stocks were used: v;; UAS–Spn27A-shRNA, v;; UAS–Rh3-shRNA (gifts from N. Perrimon, L. Perkins and the Transgenic RNAi Project) and w;; sqh::GFP; mat15-GAL4 Gap43::mCherry (generated using stocks previously described and from Bloomington). The Rh3 sh-RNA was used as a control (*Ctr-RNAi*) because Rh3 is not expressed in the *Drosophila* early embryo. For gene knockdowns in follicle cells, the following stocks were used: w; sqh::GFP; UAS–Fat2-RNAi (generated for this study, UAS–Fat2-RNAi is a gift from S. Horne-Badovinac), w; Traffic jam-GAL4; Gap43::mCherry and w; Traffic jam-GAL4; Gap43::mCherry H2A::GFP (generated using stocks previously described and from Bloomington). To visualize ROCK in live and fixed embryos, we used either the kinase-dead ROCK allele fused to GFP, sqhP–GFP::ROCK(K116A) (Fig. 3b and Supplementary Fig. 4c, gift from J. Zallen, S. Simões, and R. Fernandez-Gonzalez[65]), or a wild-type ROCK allele fused to GFP, ubiP–GFP::ROCK (Supplementary Fig. 4b,d gift from Y. Bellaïche[66]). The following stocks were generated: (v)/(w); sqh::mCherry sqhP–GFP::ROCK(K116A); UAS–Spn27A-shRNA and (v)/(w); sqh::mCherry sqhP–GFP::ROCK(K116A); mat15-GAL4, w ubiP–GFP::ROCK;; mat15-GAL4 and w ubiP–GFP::ROCK;; mat15-GAL4 Gap43::mCherry. Note that knockdown conditions gave rise to phenotypes with variable expressivity, and therefore we excluded from our analysis, embryos that were not ventralized up to half of the embryo circumference in *Spn27A-RNAi* conditions and embryos that had an aspect ratio greater than 1.5 in *Fat2-RNAi* conditions

**Live and fixed imaging.** For live imaging, embryos were dechorionated with 50% bleach, then washed with water and mounted, ventral side up, onto a slide coated with glue. No. 1.5 coverslips were used as spacers to create a chamber for the mounted embryo. Because *Fat2-RNAi* embryos were wider around the circumference, we used three pieces of double-sided tape as spacers for these embryos. The chamber was filled with Halocarbon 27 oil (Sigma). Embryos were not compressed. All imaging was performed at room temperature ($\sim$23 °C).

For cross-section images, embryos were dechorionated in 50% bleach, then heat fixed (10 s in boiling 0.4% NaCl 0.03% Triton X-100), and devitellinized by vortexing in 1:1 methanol/heptane solution. Embryos were post-fixed for 30 min in 4% paraformaldehyde after immunostaining and then placed in a drop of mounting medium and sliced using a sharp scalpel blade. Embryo slices were mounted flat on a slide using AquaPolymount (Polysciences).

For all other immunostainning, embryos were fixed in 4% paraformaldehyde (20–30 min in 0.1 M phosphate buffer at pH 7.4 with 50% heptane) and manually devitellinized. After immunostaining, embryos were mounted using AquaPolymount (Polysciences). Endogenous sqh::GFP fluorescence was used to visualize myosin and AlexaFluor568 phalloidin (Invitrogen) was used to visualize F-actin. The following antibodies were used: Snail (Rabbit, 1:500, a gift from M. Biggin), Neurotactin (Mouse, 1:100, Developmental Studies Hybridoma Bank, Hybridoma Product BP 106), Armadillo (Mouse, 1:500, Developmental Studies Hybridoma Bank, Hybridoma Product N2 7A1), E-Cadherin2 (Rat, 1:50, Developmental Studies Hybridoma Bank, Hybridoma Product DCAD2) and MBS (Rabbit, 1:500, a gift from C. Tan). Secondary antibodies used were AlexaFluor568 or 647 (Invitrogen), diluted at 1:500. All images were acquired on a Zeiss LSM 710 confocal microscope, with a 40 × /1.2 Apochromat water objective or a 63 × /1.4 Planapo APO oil objective (Zeiss), an argon ion, 561 nm diode, 594 nm HeNe and 633 nm HeNe lasers. Pinhole was set between 1 and 2 Airy Units for all images.

**Laser ablation.** Laser ablations were performed using a 2-photon Mai-Tai laser set to 800 nm on a LSM710 confocal microscope (Zeiss) through a 40 × /1.1 objective, using the Zen software (Zeiss). For ablations, laser power was set at 20%, with a scan speed within the region of interest of 0.08 ms per pixel.

Line ablations: In Figs 1c,d and 6c,b, ablations were performed along a 30 μm long line (1 pixel width). For every time point (the time resolution was 1–1.5 s for VF movies of *Ctr-RNAi* and *Spn27-RNAi*, and 2.5–3 s for VF movies of *Fat2-RNAi* and for PMG movies due to higher surface curvature requiring deeper z-stacks), an ellipse was manually drawn in Fiji, using the sqh::GFP signal to visualize the opening of the apical actomyosin meshwork. Displacement after cutting was measured by dividing the ellipse's minor axis by 2. The initial recoil velocity (v0) was calculated by fitting the displacements curves with a Kelvin–Voigt exponential and taking the first derivative.

Circular ablations: In Supplementary Fig. 2, ablations were performed within a circular region of interest of 15 μm². The shape of the wound 20 s after ablation was approximated by manually drawing an ellipse in Fiji using the sqh::GFP signal to visualize the opening of the apical actomyosin meshwork. The ellipse aspect ratio and angle between the ellipse major axis and the embryo a-p axis were calculated in Fiji. For repetitive micropatterned ablations (Fig. 4), repetitive ablations (every 30–60 s) were performed along 60 or 50 μm long lines (1 pixel width).

**RNAi injection.** Arm dsRNA were generated as previously described[17] using the following primers : Arm-F: 5′-TAATACGACTCACTATAGGGAGACCACCCTG GTTACCATAGGCCAGA-3′; Arm-R: 5′-TAATACGACTCACTATAGGGAGA CCACTGCCATCTCTAACAGCAACG-3′ (primers included the sequence of the T7 promoter); newly laid embryos were injected laterally and incubated for 2.5–3 h at room temperature (23 °C) before imaging gastrulation.

RNA *in situ* hybridizations were performed using standard procedures. Embryos were fixed twice in 4% paraformaldehyde and hybridized with Dig-labeled RNA probes (Roche). Alkaline phosphatase was used to detect Dig probes, and embryos were stained with NBT/BCIP solution (Roche). Stained embryos were observed using a 20 × /0.50 Nomarski objective on a AX10 microscope (Zeiss).

**Image processing and analysis.** Images were processed using Fiji (http://fiji.sc/ wiki/index.php/Fiji) and MATLAB (MathWorks).

A Gaussian filter (kernel = 3 pixels, σ = 0.5–1) was applied to images. Apical images (myosin, ROCK, MBS or actin signals) are maximum intensity projections of 2–5 μm. Membrane images are single sections 1–2 μm below the apical projection. We used custom MATLAB software, Embryo Development Geometry Explorer (EDGE)[67], to segment images for quantification of apical area and fluorescent signals intensities. EDGE automatically segmented cell membranes and we manually corrected for any error in the segmentation. Because the apical surfaces of *Fat2-RNAi* embryos are curved we first applied a 'flattening' method before processing the images using EDGE: the max myosin signal intensity in z was used to generate a rough map of the embryo surface. A Fourier transform was used to generate a smooth continuous surface.

We selected Gap43::mCherry signal 2 μm below the peak myosin intensity to estimate apical cell shape. To calculate cell anisotropy apical cell outlines were fitted with ellipses and major axis length and minor axis length were measured, as well as the orientation of the ellipse relative to the embryonic a-p axis. Cell anisotropy relative to the embryonic axis was obtained by dividing the orthogonal projection of ellipse axis length along the a-p axis by the orthogonal projection of ellipse axis length along the d-v axis for individual cells[17]. Myosin intensity in individual cells was quantified as follows: we smoothed sqh::GFP images using a Gaussian smoothing filter (kernel = 3 pixels, σ = 0.5), and clipped intensity values 2.5 s.d. above the mean to substract background of cytoplasmic myosin. We then made

maximum-intensity $z$ projections of myosin (averaging the three highest-intensity values) and integrated the intensity of all the pixels in a given cell. ROCK intensity in individual cells was quantified using the same strategy except that we did not clip any intensity values (we do not detect a high cytoplasmic ROCK background with either sqh$^P$–GFP::ROCK(K116A) or ubi$^P$–GFP::ROCK constructs).

To analyse early myosin pulsing behaviour, we calculated the cross-correlation between a myosin pulse (or coalescence) and the change in apical surface area (defined as the constriction rate). Temporal cross-correlation was determined using the 'crosscorr' function in MATLAB. To calculate cross-correlation between constriction rate and myosin rate of change in *Spn27A-RNAi* and *Ctr-RNAi* embryos, we first smoothed apical area and myosin intensity curves by averaging values at a given time point with immediately neighbouring time points (3 time points total). We then calculated a rate for each time point by subtracting the value following a time point by the preceding time point and dividing by the time interval. Pearson's correlations were calculated using 40 time points for each cell. To compare the frequency of instances of myosin pulses or rapid area reduction between embryos, we used methods previously described[36]. In brief, myosin signal and apical area cures were smoothed and we calculated instantaneous rates of change for each time point. Rapid myosin increase or rapid area reductions were defined as instances where the rate exceeded a threshold of one s.d. above the mean rate for all wild-type cells.

To quantify the relative position of different proteins in the myosin regulatory pathway, we measured the radial intensity profile of different proteins relative to the centre of mass of ROCK signal. We applied an edge erosion of 3 pixels on segmented images. ROCK-GFP signal was thresholded (from 60 to 90%) and the weighted centre of intensity was calculated for the thresholded image for every cell. Radial intensity distribution for a protein in a given cell was then obtained by averaging the intensity profiles along 32 radial lines emanating from the ROCK centre of mass towards the cell outline. The intensity profiles were calculated from smoothed images using a Gaussian smoothing filter (kernel = 3 pixels, σ = 0.5–0.7). To quantify and classify different types of ROCK organization within the cells, we developed a metric based on ROCK intensity profiles that distinguished three groups corresponding to condensed focus, diffuse ROCK, or rings. In control conditions, every cell falls into the first category (ROCK condensed foci). Note that in Fig. 3d and Supplementary Fig. 4b, the plots represent the subpopulation of the *Spn27A-RNAi* cells where ROCK is clearly organized into rings, illustrating the relative positions of different proteins with respect to the centre of ROCK. The total repartition of the analysed cells indicating the different configurations of the cytoskeleton is given in Supplementary Fig. 4c,d.

**Statistics.** Statistical analyses were performed using the MATLAB statistics toolbox. Normality of the distributions was first assessed using a one-sample *t*-test; *P* values were then calculated using a two-tailed unpaired *t*-test.

**Data availability.** The authors declare that all data supporting the findings of this study are available within the article and its Supplementary Information files or from the corresponding author on request.

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

## Acknowledgements

We thank S. Horne-Badovinac (University of Chicago, Chicago, IL, USA), N. Perrimon (Harvard University, Boston, MA, USA), J. Zallen (Sloan Kettering Institute, New York, NY, USA), Y. Bellaïche (Insitut Curie, Paris, France), M. Biggin (Lawrence Berkeley National Laboratory, Berkeley, CA, USA), S. Wasserman (University of California, San Diego, CA, USA), the Bloomington *Drosophila* Stock Centre and the Developmental Studies Hybridoma Bank for kindly providing flies and antibodies used in this study. We thank I. Cheeseman, F. Schweisguth and all laboratory members for discussion and helpful comments on this manuscript. This work was financially supported by Grants R01GM105984 from the National Institute of General Medical Sciences and Grant 14GRNT18880059 from the American Heart Association to A.C.M., an EMBO fellowship ALTF 1082-2012 to S.C., a Grant DMS-1219754 from the National Science Foundation to B.E. and Grants R01 HD044750 and R21 ES019259 from the National Institutes of Health and National Science Foundation CAREER IOS-0845775 and CMMI-1100515 to L.A.D. C.J.M. also had support from a National Institutes of Health Biomechanics in Regenerative Medicine Training Grant 2T32EB003392. Any opinions, findings and conclusions or recommendations expressed in this material are those of the authors and do not necessarily reflect the views of the NIH or NSF.

## Author contributions

S.C. and A.C.M. designed the project, analysed the data and wrote the paper. S.C. performed the experiments. C.J.M., B.E. and L.A.D. developed the *in silico* model. E.D.V. developed the method to quantify radial intensity profiles.

## Additional information

**Competing interests:** The authors declare no competing financial interests.

