## [Peer Review File · Nature Communications]

Reviewers' comments:

Reviewer #1 (Remarks to the Author):

In this manuscript, Chanet and colleagues use biophysical approaches to investigate the role of geometric and mechanical constraints in orienting actomyosin networks and contractile forces. The authors previously showed that during mesoderm internalization in *Drosophila*, tension across the mesoderm is anisotropic and greater along the anterior-posterior axis of the animal (the long axis) than along the dorsal-ventral axis (the short axis). Here, they use RNAi-based manipulations, together with laser ablation or dsRNA-mediated weakening of cell junctions, to show that increasing the length of the short axis or decreasing the length of the long axis results in isotropic tissue tension in the mesoderm. Changes in tissue geometry lead to slower, incomplete, and isotropic cell constriction. Weakening junctions to isolate cells from their neighbors rescued the speed and extent of apical constriction in embryos with isotropic mesoderm geometry. The authors find that changes to the geometrical constraints sustained by the mesoderm affect the architecture of actin, myosin and upstream regulators of myosin, which form rings that do not exist in the wild type. Using laser cuts in embryos with an isotropic mesoderm to remove lateral constraints rescued anisotropic constriction and the formation of myosin fibers. Similarly, removing anterior and posterior constraints in the wild type resulted in myosin ring formation and isotropic constriction. A computational model showed that anisotropies in the stiffness of the boundaries where actin filaments are anchored can lead to directional force generation along the axis of maximum resistance and actomyosin fiber formation. Interestingly, the model predicts that myosin motors align with the axis of maximum stiffness, as it is easier for them to move along the softer axis. Finally, the authors show that the internalization of the posterior midgut, which is driven by the same signaling pathway than mesoderm invagination, involves isotropic cell constriction mediated by myosin rings. The authors propose that geometry-imposed mechanical constraints can tune the output of actomyosin networks regulated by similar biochemical programs.

The manuscript is very nicely written, and the data are clear and well documented. I found some of the experimental approaches really original (e.g. the RNAi-based changes in mesoderm geometry). The computational model not only helps with the interpretation of the data, but makes valuable predictions, such as the alignment of the myosin motors, that would have been difficult to obtain from experiments (unfortunately, they are also difficult to test!). I just have a few questions about data interpretation and a couple of suggested additional controls to solidify the model presented in the paper.

MAJOR POINTS

1. From Figure 1e, it seems like the incidence of tissue tears upon arm RNAi injection is greater in the RNAi embryos with disrupted mesoderm geometries (*Spn27A* and *Fat2*) than in the wild type? Is that the case? Based on Figure 1d, and assuming no difference in drag between wild-type and RNAi embryos, it seems that actomyosin networks display reduced contractility in the RNAi embryos, which would be consistent with fewer tears when adhesion is reduced. Please clarify.
2. The authors are proposing that isotropic mechanical constraints (or no constraints) result in the cup-like invagination of the posterior midgut (PMG). A prediction of this model is that repetitive line ablation experiments such as those in Figure 4 should be able to cause anisotropic constriction of PMG cells and possibly the formation of a furrow. This point should be tested.
3. An important point that is missing is whether there are any functional consequences of the isotropic contraction of mesoderm precursors in the invagination of the mesoderm? From the movies, it seems like at least in *Fat2*-RNAi embryos the ventral furrow forms. Do these embryos develop? Until what stage? If they do not hatch, is it because defects in the mesoderm, or are there additional defects that cause the lethality?
4. The authors propose that actomyosin network architecture may also depend on cell shape. On

this note, it would be useful to know what happens first, the formation of myosin fibers or the increase in cell shape anisotropy. In addition, the authors should use their computational model to test potential "confinement" effects caused by changes in cell shape without having to invoke anisotropic stiffness.

MINOR POINTS

1. The authors argue that pulsatile dynamics are similar in wild-type and RNAi embryos (page 9). However, this is not shown. What are the period and amplitude of pulsation in RNAi embryos compared to the wild type? Could there be effects of the different actomyosin network architectures on the pulsatile activity? Could differences in pulsatile activity result in the slower and stalled constriction observed in the RNAi embryos?
2. Are there pulsatile behaviours during the ingression of the endoderm? If so, are pulse amplitude and period similar to those found in the mesoderm of RNAi embryos where cells also undergo isotropic pulsing?
3. How do Rok/myosin/actin rings form in RNAi embryos? In other words, which molecule is being affected by the mechanical constraints imposed by the tissue? The authors should disrupt each one of these molecules and determine which one prevents ring formation.
4. In the Methods, the authors mention that a flattening method was applied to images from Fat2-RNAi embryos prior to processing. The method should be described, or if it already is – did they refer to the orthogonal projections mentioned below? – then it should be clearly indicated.

TYPOS

1. Page 6, paragraph 2: "followed the displacement" should be "and followed the displacement".
2. The authors indicate that they used the "crosscoef" function in Matlab to calculate cross-correlations. However, I could not find any such function in Matlab. Do they refer to "corrcoef"?
3. Figure 5f: "motors alignment" should be "motor alignment".

Reviewer #2 (Remarks to the Author):

Review of manuscript entitled
“Actomyosin meshwork mechanosensing enables tissue shape to orient cell force”
by Soline Chanet et al.

Summary

This manuscript aims at demonstrating how, complementary to the biochemical activity pattern governed by genetic expression, the global mechanical balance (and thus the geometry) of tissue mediates phenotypical changes in cells, namely cell morphology and actomyosin cytoskeleton arrangement. Experiments where either the geometry of the tissues or the patterning of activity differ in a controlled manner show that these tissue-scale changes feed back on cell phenotype far away from the areas that have been perturbed. The original phenotype is rescued when mechanical balance is restored using laser cuts or tissue tearing, showing that these phenotypical changes are indeed caused by alterations of the tension felt by cells. Molecular dynamics simulations propose a plausible mechanism to explain phenotypical changes of the actin and myosin within cells subjected to different mechanical conditions. Since actomyosin is known to be causal of the appearance of tension and/or morphogenetic movements in the tissues studied here (*Drosophila* mesoderm and post-mid gut endoderm), the authors link the observed tension or movement with the actomyosin organisation.

Comments

Phenotypic changes of cells resulting from global mechanical balance

The experimental work described in the manuscript gives compelling evidence that a change in the geometry of either the tissue (or assembly of tissues) or the patterning of myosin activation gives rise to global changes in the mechanical tension, and, in turn, in the cell shape and actomyosin organisation.

This is a novel and striking result, which brings a new perspective to recent work by several groups in the *Drosophila* community showing how global mechanical balance affects local processes such as cell intercalation (Collinet et al, NCB 2015, Ref. 61; Etournay et al, eLife 4:e07090, 2015, unreferenced in manuscript; I believe that both these references should be acknowledged for showing this). In these papers however, consequences of global tension alteration were shown on tissue dynamics at the scale of a few cells and not in the cytoskeletal arrangement, to the difference of the case of the mesoderm during VF formation studied here.

Further, using laser cuts or the tears that occur spontaneously in *arm-RNAi* embryos, the authors make clear that the way that the geometry entails these local phenotypical changes is through a modulated mechanical resistance to deformation of the neighboring tissues, or, in mechanical terms, by a change of the boundary conditions.

Mechanosensing

Beyond the phenotypical change, the authors question how the mechanical signal resulting from a geometrical change feeds back on the stress generation by actomyosin. They show that it is not necessary to consider a transduction step, in which the mechanical state of cells would entail a specific biochemical cascade leading to a modified phenotype in terms of proteins expressed, but that the self-organisation of actomyosin

under different mechanical boundary conditions is *per se* sufficient to entail a different mechanical response.

The authors show in molecular dynamics (MD) simulations that the boundary stiffness feeds back both on the mechanical balance (the force exerted by a model network of actomyosin on its boundaries increases with environment stiffness) and actomyosin organisation and cell shape. This result has already been obtained by Borau et al. (PLoS one 7:e49174, 2012), Parameswaran et al. (J. Appl. Physiol., 116:825) using MD simulations (with some differences with the present MD simulations, Borau et al have Bell's rate for detachment of crosslinkers, both have a stall force for myosin and no actin turnover), and Etienne et al. (PNAS 112:2740, 2015, who have a continuum model derived from similar assumptions as the author's). These results should be presented and compared to the present ones, since they go in deeper details on the mechanism and the shape of the stiffness-dependence of the force exerted by the cell on its environment.

These papers make clear that the geometric phenotype (cell shape and acto-myosin organisation) and the force exerted by the cell on its environment are not to be considered separately as one being the result of the other, but are the two unseparable results of the mechanical balance that arises for a given boundary condition and myosin activation: this is indeed a well established concept in physics, that stress and strain are related by a rheological equation, which is a property of the material, but neither can be determined without considering the mechanical balance (and thus boundary conditions) that will give the set-point in this stress-strain relation. There is no disagreement here with the approach of the authors to model the actomyosin network behavior in presence of anisotropic boundary conditions, since they define a rheology (via the rules of interaction of their model actin and myosin filaments) and boundary conditions (via their rules of attachment-detachment and stiffness of boundaries). However, the authors do not sufficiently question their numerical results in order to support their claim that actomyosin enhances its action in a given direction. Indeed, the papers mentioned above clearly show that stress will grow with environment stiffness (even if isotropic). The manuscript does not make clear whether with anisotropic boundary conditions there is an enhancement of the stress in addition to this rigidity-sensitive behavior. Namely, the authors should (a) measure the stress that their MD simulation predicts for low and high stiffness in isotropic conditions, (b) compare with the stress in an MD simulation with anisotropic environment stiffness, so as to quantify whether there is an increase of stress along the stiff direction beyond the one expected from (a). Depending on the results, the meaning of the phrase "orient cell force" should be clarified as to whether the stress is anisotropic because boundary conditions are, or whether there is a further enhancement of this anisotropy due to a reorganisation.

This leads to a further question on the dynamics of establishment of this anisotropy. The dynamics of the actomyosin structures is described in the MD simulations, but not *in vivo*. In particular, A-P structures in the mesoderm (Fig 3a) do not seem necessary for anisotropic tension, as they begin to appear toward the end of the laser cut experiment. Would the authors be able to characterise the dynamics either of the reorganisation or of the stress anisotropy? Even if not, I believe the question of the timings must still be mentioned in the main text.

Invagination

The authors highlight that the cup-like invagination of PMG is associated with isotropic tension and actomyosin rings, while furrow-like mesoderm invagination is associated with anisotropic tension and actomyosin longitudinal fiber-like structures. Since they are able to disrupt this latter organisation in RNA assays, it would be of the utmost interest to give some account of the resulting phenotypes in terms of shape of the invagination for these two genetic perturbations.

Recommendation

Overall, the manuscript is of extremely high quality both in the research it presents and the way it presents it. I am of the opinion that this manuscript will be a paper of reference in further work on the interplay of global mechanical equilibrium and genetic patterning in morphogenesis, and it is important that the more subtle details of this interplay are fully acknowledged. Thus I recommend a manuscript revision that would discuss these subtler details and possibly include some more quantification of the experimental results presented in this manuscript.

Molecular dynamics simulations. The appearance of a self-organised ring in the simulations is surprising to me, and was not reported in previous similar models (see references above). Some description of the mechanism would be welcome, as well as a test of its dependence on parameters other than the stiffness (e.g., width of the "boundary region", currently 0.5 μm , but also filament and motor density).

Minor comments

Curvature. Pages 5–6, the influence of genetic modifications on the geometry of myosin pattern are discussed, however they also change the curvature of the epithelium, this may be worth mentioning.

F-Actin. Page 8, the authors write "F-actin organization mirrored that of myosin, becoming depleted from the apical center and organized into rings centered within the apical domain (Supplementary Fig. 4a, b)." According to Supp. Fig 4b, in perturbed conditions actin is farther from center than myosin is, and in panel a it is not clear to me that it is structured in rings and that these are away from the cell-cell junctions.

Mechanosensing literature. The authors highlight that the sensing of mechanical conditions in their experiments can be explained without resorting to a transduction step that, however they do not cite the previous literature on this mechanism. In addition to the modeling papers mentioned above, I can suggest experimental work of Mitrossilis et al., PNAS 106:18243, 2009, Trichet et al., PNAS 109:6933, 2012; and modeling work of Zemel et al., J. Phys.: Condens. Matter, 22:194110, 2010, Marcq et al., Biophys. J., 101:L33 2011.

Perijunctional actomyosin. In the end of the discussion, the authors oppose the versatility of medio-apical actin network (studied here) to perijunction actomyosin. However they do not support the claim that perijunctional myosin would not have a similar

(although 1D) mechanosensitivity, and indeed Fernandez et al., Dev. Cell, 17:736 have shown some mechanosensitive changes in 1D junctional actomyosin cables.

Laser ablation methods. Page 19, I have difficulty to make out which part of the description applies to straight line and which to circular ablations.

Molecular dynamics simulations.

In addition to the above comment on ring formation, I believe the simulations would be made more convincing by providing some analyses of the robustness of their results with respect to variations in two classes of parameters : first, the dynamic ones (viscosity, rates of diffusion,...) which are not at all discussed and are (I believe) thought not to have a first order influence, and second on filament and motor numbers.

The measure of anisotropy in terms of a "maximum" and "minimum force" are unclear to me, is an average force not relevant? If so, why, and would not an increase in filament density allow to use an average?

It seems to me not obvious (if not impossible) to create a homogeneous random network of uniform finite length fibers within a finite domain, either the filaments will be aligned with boundaries or their density will be less close to the boundary (the so-called effect of excluded volume). Could the authors document how they initialise their simulation?

Reviewer #3 (Remarks to the Author):

This manuscript reports on the impact of tissue geometry on actomyosin organization and on the directionality of cell generated forces. To address this question, the authors use the early *Drosophila* embryo as a model system. During gastrulation, an epithelial furrow forms by contraction of a narrow patch of cells. Using genetic and mechanical perturbations, the author show that the geometry of this patch determines the orientation of actomyosin networks and thus, the anisotropy of mechanical stress at the tissue level.

The question is topical and some of the data are very interesting and convincing. However I consider that the data do not support the main conclusion of the manuscript, which is summarized in the title 'actomyosin networks are mechanosensing' enabling 'tissue to orient cell forces.'

The convincing and very nice part of this manuscript focuses on the effects of a contractile tissue shape onto force generation (Fig 1, 2 and 6). It extends previous observation and suggestions of the same group (Martin et al, 2010) that anisotropy of cell shape changes depends on the geometry of the contractile ventral furrow primordium. This hypothesis was also tested by optogenetic control by Guglielmi et al (2015). Here the authors convincingly show that the geometry of the primordium also determines the anisotropy of stress.

I don't think that the authors provide evidence that mechanical 'constraints' such as stiffness at the boundary of the contractile tissue (or cells) alter the actomyosin organization and thus force direction. Cells are stretched along the long axis of the contractile tissue and it is likely that the actomyosin network reorganizes accordingly. The fig. 3a (and 3c) mainly show that the actomyosin network organization anisotropy follows the anisotropy in cell shape. In Fig 3a. the actomyosin network can be seen as "fibers" in squeezed cells but also as rings in more isotropic cells.

As a consequence, while interesting, a cellular model assuming that the cell boundary has an anisotropic resistance (or stiffness) has little relevance to the present study. Anisotropic spatial confinement has been shown to organize actin networks in vitro (Marina Soares e Silva, *Soft Matter*, 2011). Alternative explanations are likely to be more justified than the one proposed by the authors : the observed actomyosin organization could be induced by the stretching of the cells along the a-p axis in the contractile tissue. The authors only allude to this possibility in their discussion (and references 2 & 3). I would expect the authors explore this mechanism thoroughly. If not, they have to test their model through a series of new experiments, in which stiffness at tissue boundaries would be modified (independent of cell shape).

The manuscript concludes that the actomyosin meshworks adapt to external mechanical pattern and redirect cell forces. However the stress anisotropy comes from the very cells which are contractile in the furrow primordium. I don't see the external mechanical pattern and at this stage, the conclusions drawn by the authors are not supported by experimental evidence.

We would like to thank all the reviewers for their helpful and constructive comments. We hope the reviewers will be enthusiastic about the changes we made in our manuscript.

Reviewer #1 (Remarks to the Author):

In this manuscript, Chanet and colleagues use biophysical approaches to investigate the role of geometric and mechanical constraints in orienting actomyosin networks and contractile forces. The authors previously showed that during mesoderm internalization in *Drosophila*, tension across the mesoderm is anisotropic and greater along the anterior-posterior axis of the animal (the long axis) than along the dorsal-ventral axis (the short axis). Here, they use RNAi-based manipulations, together with laser ablation or dsRNA-mediated weakening of cell junctions, to show that increasing the length of the short axis or decreasing the length of the long axis results in isotropic tissue tension in the mesoderm. Changes in tissue geometry lead to slower, incomplete, and isotropic cell constriction. Weakening junctions to isolate cells from their neighbors rescued the speed and extent of apical constriction in embryos with isotropic mesoderm geometry. The authors find that changes to the geometrical constraints sustained by the mesoderm affect the architecture of actin, myosin and upstream regulators of myosin, which form rings that do not exist in the wild type. Using laser cuts in embryos with an isotropic mesoderm to remove lateral constraints rescued anisotropic constriction and the formation of myosin fibers. Similarly, removing anterior and posterior constraints in the wild type resulted in myosin ring formation and isotropic constriction. A computational model showed that anisotropies in the stiffness of the boundaries where actin filaments are anchored can lead to directional force generation along the axis of maximum resistance and actomyosin fiber formation. Interestingly, the model predicts that myosin motors align with the axis of maximum stiffness, as it is easier for them to move along the softer axis. Finally, the authors show that the internalization of the posterior midgut, which is driven by the same signaling pathway than mesoderm invagination, involves isotropic cell constriction mediated by myosin rings. The authors propose that geometry-imposed mechanical constraints can tune the output of actomyosin networks regulated by similar biochemical programs.

The manuscript is very nicely written, and the data are clear and well documented. I found some of the experimental approaches really original (e.g. the RNAi-based changes in mesoderm geometry). The computational model not only helps with the interpretation of the data, but makes valuable predictions, such as the alignment of the myosin motors, that would have been difficult to obtain from experiments (unfortunately, they are also difficult to test!). I just have a few questions about data interpretation and a couple of suggested additional controls to solidify the model presented in the paper.

We thank the reviewer for their thoughtful comments and the obvious care they took to write their review. We agree that the alignment of bipolar myosin filaments is a valuable and concrete prediction that will be the next step in understanding the mechanism of this cell-level response to force.

MAJOR POINTS

1. From Figure 1e, it seems like the incidence of tissue tears upon *arm* RNAi injection is greater in the RNAi embryos with disrupted mesoderm geometries (*Spn27A* and *Fat2*) than in the wild type? Is that the case? Based on Figure 1d, and assuming no difference in drag between wild-type and RNAi embryos, it seems that actomyosin networks display reduced contractility in the RNAi embryos, which would be consistent with fewer tears when adhesion is reduced. Please clarify.

The reviewer is correct, we observed a higher incidence of tissue tears after *arm*-RNAi injections in *Spn27A*-RNAi (n=26 tears, 3 embryos) or *Fat2*-RNAi (n=22 tears, 3 embryos) embryos than in *Ctrl*-RNAi embryos (n=13 tears, 3 embryos). We also agree that based on Figure 1d one can conclude that actomyosin networks display reduced tension along the anterior-posterior axis in the RNAi embryos, assuming no difference in drag. The fact that a normal constriction rate and full cell constriction occurred upon release of mechanical constraint after *arm*-RNAi injection (Figure 2e) argue for similar contractility at the cell level between wild-type, *Fat2*-RNAi and *Spn27A*-RNAi embryos. However, cells experience different mechanical contexts in wild-type and RNAi embryos.

The higher number of tears seen in *Spn27A-RNAi* embryos can be explained by the expanded ventral domain and the increased surface area of contracting tissue. *Fat2-RNAi* embryos do not have greater surface area, but invagination is delayed and there is more time for tears to develop. Because what determines the number of tears in the ventral tissue is unclear, but the rationale for orientation is, we focus on orientation in the text.

2. The authors are proposing that isotropic mechanical constraints (or no constraints) result in the cup-like invagination of the posterior midgut (PMG). A prediction of this model is that repetitive line ablation experiments such as those in Figure 4 should be able to cause anisotropic constriction of PMG cells and possibly the formation of a furrow. This point should be tested.

We agree that changing the mechanical pattern in the PMG would be interesting. However, this experiment is not technically feasible. The problem is that the posterior part of the embryo is highly curved and confocal microscopy is planar. We have tried to perform line ablations on similarly curved *Fat2-RNAi* embryos. Instead of cutting a line, the laser makes 2 spot ablations where the laser is in focus with the embryo surface. We have multiple lines of evidence that suggest that cells respond to the mechanical constraints: 1) Changing the mechanical context of the tissue with *Spn27A-RNAi* results in myosin rings, similar to the PMG. In addition, *Spn27A-RNAi* embryos often exhibit cup-like indentations, like those in Figure S1 – *sim* in situ. We have added this point to the text (p.14). 2) We have performed a mechanical suppression of the *Spn27A-RNAi* phenotype, where these line ablation experiments are most cleanly done (Figure 4a-d). 3) We can promote myosin ring formation by making vertical line ablations in a wild-type embryo (Figure 4f-i).

3. An important point that is missing is whether there are any functional consequences of the isotropic contraction of mesoderm precursors in the invagination of the mesoderm? From the movies, it seems like at least in *Fat2-RNAi* embryos the ventral furrow forms. Do these embryos develop? Until what stage? If they do not hatch, is it because defects in the mesoderm, or are there additional defects that cause the lethality?

Amazingly, the majority of the round *Fat2-RNAi* embryos hatch and give rise to normal but smaller larvae (n=88/100 *Fat2-RNAi* embryos), indicating that the presumptive mesoderm invaginates and the round shape of the embryo does not prevent further embryogenesis. In comparison 97% of wild-type embryos hatched (n=100 wt embryos). However, in some instances (8%, 3/35 *Fat2-RNAi* embryos), round embryos failed to form a furrow and to invaginate the presumptive mesoderm. *Spn27A-RNAi* results in more severe defects: 31% (12/39) *Spn27A-RNAi* embryos failed to invaginate an a-p oriented furrow. We have now included examples of *Spn27A-RNAi* and *Fat2-RNAi* that do not invaginate as Video 2.

4. The authors propose that actomyosin network architecture may also depend on cell shape. On this note, it would be useful to know what happens first, the formation of myosin fibers or the increase in cell shape anisotropy. In addition, the authors should use their computational model to test potential “confinement” effects caused by changes in cell shape without having to invoke anisotropic stiffness.

There is a precedent for cell shape influencing the actin cytoskeleton and we mentioned it in the Discussion. We would like to highlight however that the formation of oriented myosin fibers occur before the cells have fully constricted along the d-v axis, suggesting that fiber formation precedes a fully anisotropic cell shape. We have now added a supplementary movie (Video 5) to make this point. Our *in silico* model suggests a mechanism that involves how motors and filaments move during contraction, but does not involve confinement. It does not exclude the effects of cell shape but rather illustrates that compliance of the surrounding tissues can also influence the cytoskeleton by influencing how it contracts. The advantage of our modeling framework is that it allows us to isolate the contribution of tissue compliance from that of cell shape. We have tested in our model the effect of confinement by embedding our simulation in an ellipse, mimicking anisotropic cells shape (see Rebuttal Figure 1). Interestingly, confinement does not explain the effects we see *in vivo*. The averaged force produced within an ellipse is perpendicular to the long axis (Rebuttal Figure 1), whereas we measured *in vivo* orientation of force production along the axis of greater resistance. Because it is how the network contracts, not

“confinement”, that best explains our data, we have focused on explaining this mechanism in the text.

Rebuttal Figure 1: Anisotropic cell shape do not orient forces along the long axis.

To test confinement on force production, the simulations were embedded within an ellipse (ellipse major radius=2 μm ; ellipse minor radius=1 μm). The rose diagram illustrates the average force generated on boundary springs for 10 simulations, circles divisions on the rose diagrams represent 10 nN.

MINOR POINTS

1. The authors argue that pulsatile dynamics are similar in wild-type and RNAi embryos (page 9). However, this is not shown. What are the period and amplitude of pulsation in RNAi embryos compared to the wild type? Could there be effects of the different actomyosin network architectures on the pulsatile activity? Could differences in pulsatile activity result in the slower and stalled constriction observed in the RNAi embryos?

We thank the reviewer for this comment. We have shown in multiple ways that control and RNAi embryos all display pulsatile behavior: 1) we have provided for each condition a trace of myosin intensity and apical area for one representative cell (Supplementary Figure 5b); 2) we showed the cross-correlation between constriction rate and change in myosin intensity (Supplementary Figure 5c), which is high if pulsing is present, but low when pulsing is disrupted (Vasquez et al., JCB, 2014); 3) To further compare pulsatile dynamics between control and *Spn27A-RNAi* embryos, we have now calculated the frequency of events involving rapid myosin intensity increase as well as the frequency of rapid area reduction and found no significant difference between control and *Spn27A-RNAi* cells (Supplementary Figure 5d). It is not possible to perform a quantitatively comparable analysis of pulse frequency for *Fat2-RNAi* cells because the greater curvature of these embryos necessitates a deeper z-stack and thus very different time-resolution. We have shown in Supplementary Figure 5 more qualitative examples of pulsing in *Fat2-RNAi* cells and the correlation of apical cell area reduction with myosin intensity change.

With regards to pulse amplitude: We have shown that the magnitude of apical constriction is associated with persistence of myosin intensity and structure (Mason et al., JCB, 2016). Thus, apical constriction is most closely related to the overall magnitude of apical myosin intensity and we have shown that myosin intensity is similar between wild-type, *Spn27A-RNAi*, and *Fat2-RNAi* embryos. We think that this is the most compelling way to compare myosin levels and thus contractility between strains; we show this in Supplementary Figure 5a. Note that pulsing *per se* is not required for apical constriction (Vasquez et al., JCB, 2014).

2. Are there pulsatile behaviours during the ingression of the endoderm? If so, are pulse amplitude and period similar to those found in the mesoderm of RNAi embryos where cells also undergo isotropic pulsing?

The PMG cells also display pulsatile behavior. We have added data showing the pulsatile behavior of myosin in PMG and the cross-correlation showing that a pulse of myosin correlates with a rapid change in the apical cell area. These results are now shown in Supplementary Figure S7. Because the PMG is on the end of the embryo, and folds in such a way that makes cells difficult to follow, our quantitative analysis of pulsing frequency is not comparable between the PMG and ventral furrow. The exact pulsing frequency of the PMG is not a critical point for the paper. They pulse in a way that qualitatively resembles the ventral furrow, but myosin ends up as rings, similar to *Spn27A-RNAi* ventral cells.

3. How do Rok/myosin/actin rings form in RNAi embryos? In other words, which molecule is being affected by the mechanical constraints imposed by the tissue? The authors should disrupt each one of these

molecules and determine which one prevents ring formation.

This is an interesting question and one that will probably take decades to solve. The problem is that it is practically impossible to test the individual roles of Rok, actin or myosin *in vivo*; they are all interdependent. Rok kinase activity is required for myosin localization, but also affects its own localization: Disrupting Rho-kinase results in the immediate (~ 10 sec) loss of myosin and disruption of its own localization to the apical center (Coravos and Martin, *Dev. Cell*, 2016). F-actin regulates Rok localization: depolymerizing F-actin, or depleting the formin *diaphanous* results in a loss of Rho-kinase localization (Coravos and Martin, *Dev. Cell*, 2016). In addition, disrupting actin polymerization or turnover disrupts the actin network's attachment to the junction (Mason et al., 2013; Jodoin et al., 2015). Thus, all of these systems are interdependent. This is why we have relied on modeling to understand the collective properties of such a system. Our *in silico* model predict a myosin-depend reorientation of actin filament in response to anisotropic boundary stiffness, a mechanism that does not invoke any upstream biochemical signaling (like Rok). We therefore believe that Rok altered localization is a secondary consequence of the actomyosin network self-organization depending on external constraint.

We have attempted to test the role of myosin in the mechanosensing response.

To do so, we expressed a myosin light chain phospho-mutant (*sqh-TA*) *in vivo*, in a context where endogenous wild-type *sqh* is highly reduced (about 90% knock down). We know that this mutant reduces myosin motor velocity by ~ 50-70% (J. Sellers and C. Vasquez, personal communication). We found that these mutants generate significantly less tension along the a-p axis compared to wild-type motor (*sqh-TS*), and that epithelial tension was more isotropic even though the tissue shape was unchanged (Rebuttal Figure 2a, b). Remarkably, the *sqh-TA* mutant resulted in a propensity of cells to form myosin rings rather than oriented fibers with ROCK located in the center of the rings (Rebuttal Figure 2c; percentage of cells with rings: 7% in *sqh-TS*, control motor, n=183 cells, 3 embryos; 33% in *sqh-TA*, n=316 cells, 4 embryos). This suggested that robust myosin activity is required to generate the force that orients the cytoskeleton. We propose that actomyosin meshworks compose a mechanical feedback loop that generate, but also respond to tensile forces. These results are entirely consistent with our model. However, this perturbation does not separate the role of myosin in force production vs. sensing, which we think is inseparable. We have chosen not to include these results in the manuscript, in order to document fully our model, which we think provides the same insight.

Rebuttal Figure 2: ***sqh-TA* mutant motor generated lower tension across the VF than wild-type control and organized into rings.**

a. Bright-field images showing embryo shapes of wild-type, *sqh-TS* (control motor) and *sqh-TA* embryos. Scale bar, 50 μm.

b. Box-and-whisker plot of initial recoil velocity (v_0) along a-p and d-v, measured after line ablation performed in *sqh-TS* (n=15 vertical incisions and n=9 horizontal incisions), and *sqh-TA* (n=15 vertical incisions and n=10 horizontal incisions).

***, $p < 0.001$, *, $p < 0.05$, n.s., not significant.

c. Images of VF cells in fixed *sqh-TS* and *sqh-TA* embryos showing the localization of ROCK::GFP (green) and myosin (*sqh::mCherry*, magenta).

4. In the Methods, the authors mention that a flattening method was applied to images from Fat2-RNAi embryos prior to processing. The method should be described, or if it already is – did they refer to the orthogonal projections mentioned below? – then it should be clearly indicated.

We thank the reviewer for pointing this out. We have now added a description of these methods to the manuscript (Methods section p.22): Briefly, the max myosin signal intensity in z was used to generate a rough map of the embryo surface. A Fourier transform was used to generate a smooth continuous surface. We selected Gap43::mCherry signal 2 μm below the peak myosin intensity to estimate apical cell shape.

TYPOS

1. Page 6, paragraph 2: “followed the displacement” should be “and followed the displacement”.

Thank you, we have fixed this.

2. The authors indicate that they used the “crosscoef” function in Matlab to calculate cross-correlations. However, I could not find any such function in Matlab. Do they refer to “corrcoef”?

The function used is “crosscorr”, we changed this in the text (p. 23).

3. Figure 5f: “motors alignment” should be “motor alignment”.

We thank the reviewer for catching this, we fixed it.

Reviewer #2 (Remarks to the Author):

Review of manuscript entitled
'Actomyosin meshwork mechanosensing enables tissue shape to orient cell force'
by Soline Chanet et al.

Summary

This manuscript aims at demonstrating how, complementary to the biochemical activity pattern governed by genetic expression, the global mechanical balance (and thus the geometry) of tissue mediates phenotypical changes in cells, namely cell morphology and actomyosin cytoskeleton arrangement. Experiments where either the geometry of the tissues or the patterning of activity differ in a controlled manner show that these tissue-scale changes feedback on cell phenotype far away from the areas that have been perturbed. The original phenotype is rescued when mechanical balance is restored using laser cuts or tissue tearing, showing that these phenotypical changes are indeed caused by alterations of the tension felt by cells. Molecular dynamics simulations propose a plausible mechanism to explain phenotypical changes of the actin and myosin within cells subjected to different mechanical conditions. Since actomyosin is known to be causal of the appearance of tension and/or morphogenetic movements in the tissues studied here (*Drosophila* mesoderm and post-mid gut endoderm), the authors link the observed tension or movement with the actomyosin organisation.

Comments

Phenotypical changes of cells resulting from global mechanical balance

The experimental work described in the manuscript gives compelling evidence that a change in the geometry of either the tissue (or assembly of tissues) or the patterning of myosin activation gives rise to global changes in the mechanical tension, and, in turn, in the cell shape and actomyosin organisation.

This is a novel and striking result, which brings a new perspective to recent work by several groups in the *Drosophila* community showing how global mechanical balance affects local processes such as cell intercalation (Collinet et al., NCB, 2015, Ref. 61; Etournay et al., eLife 4:e07090, 2015, unreferenced in manuscript; I believe that both these references should be acknowledged for showing this). In these papers however, consequences of global tension alteration were shown on tissue dynamics at the scale of a few cells and not in the cytoskeletal arrangement, to the difference of the case of the mesoderm during VF formation studied here.

We thank the reviewer for acknowledging the novelty of our study. We have added the following references: Collinet et al., NCB, 2015 and Etournay et al., eLife, 2015 to the Introduction (p. 3) where we discuss various examples of tissues responding to forces. There are indeed numerous examples where mechanical forces influence cells and tissues (reviewed in Chanet and Martin, 2014). But, what we are most excited about is the striking effect that forces have on cytoskeletal arrangement in the *Drosophila* embryo.

Further, using laser cuts or the tears that occur spontaneously in *arm-RNAi* embryos, the authors make clear that the way that the geometry entails these local phenotypical changes is through a modulated mechanical resistance to deformation of the neighboring tissues, or, in mechanical terms, by a change of the boundary conditions.

Mechanosensing

Beyond the phenotypical change, the authors question how the mechanical signal resulting from a geometrical change feeds back on the stress generation by actomyosin. They show that it is not necessary to consider a transduction step, in which the mechanical state of cells would entail a specific biochemical cascade leading to a modified phenotype in terms of proteins expressed, but that the self-organisation of actomyosin under different mechanical boundary conditions is *per se* sufficient to entail a different mechanical response.

The authors show in molecular dynamics (MD) simulations that the boundary stiffness feeds back both on the mechanical balance (the force exerted by a model network of actomyosin on its boundaries increases with environment stiffness) and actomyosin organisation and cell shape. This result has already been obtained by Borau et al. (PLoS one 7:e49174, 2012), Parameswaran et al. (J. Appl. Physiol., 116:825) using MD simulations (with some differences with the present MD simulations, Borau et al have Bell's rate for detachment of crosslinkers, both have a stall force for myosin and no actin turnover), and Etienne et al. (PNAS 112:2740, 2015, who have a continuum model derived from similar assumptions as the author's). These results should be presented and compared to the present ones, since they go in deeper details on the mechanism and the shape of the stiffness-dependence of the force exerted by the cell on its environment.

Thank you for pointing out these additional studies and for the suggestion to discuss the finer details of the different models. We think this is an excellent suggestion and have now included a paragraph in the Discussion, where we have presented these additional models and have compared them to our own MD simulations. Though there are some similarities to the theoretical aspect of this study, we note that the most exciting advance of our study is to show that cells respond to mechanical constraints in a physiological process (i.e. tissue folding) and that this plays an important functional role in directing force. In addition, our experimental and theoretical analysis predicts the important role that geometry and contraction play in orienting molecular motors in a two-dimensional actomyosin meshwork that spans the apical surface of epithelial cells.

These papers make clear that the geometric phenotype (cell shape and actomyosin organisation) and the force exerted by the cell on its environment are not to be considered separately as one being the result of the other, but are the two unseparable results of the mechanical balance that arises for a given boundary condition and myosin activation: this is indeed a well established concept in physics, that stress and strain are related by a rheological equation, which is a property of the material, but neither can be determined without considering the mechanical balance (and thus boundary conditions) that will give the set-point in

this stress-strain relation. There is no disagreement here with the approach of the authors to model the actomyosin network behavior in presence of anisotropic boundary conditions, since they define a rheology (via the rules of interaction of their model actin and myosin filaments) and boundary conditions (via their rules of attachment-detachment and stiffness of boundaries). However, the authors do not sufficiently question their numerical results in order to support their claim that actomyosin enhances its action in a given direction. Indeed, the papers mentioned above clearly show that stress will grow with environment stiffness (even if isotropic). The manuscript does not make clear whether with anisotropic boundary conditions there is an enhancement of the stress in addition to this rigidity-sensitive behavior. Namely, the authors should (a) measure the stress that their MD simulation predicts for low and high stiffness in isotropic conditions, (b) compare with the stress in an MD simulation with anisotropic environment stiffness, so as to quantify whether there is an increase of stress along the stiff direction beyond the one expected from (a). Depending on the results, the meaning of the phrase "orient cell force" should be clarified as to whether the stress is anisotropic because boundary conditions are, or whether there is a further enhancement of this anisotropy due to a reorganisation.

We agree with the reviewer's synopsis of the field of rheology and the mechanics of structures such as those defined by polymer networks. To allay some of the reviewer's questions concerning the methodology we now include a copy of a manuscript that has been resubmitted to Molecular Biology of the Cell. We are confident that this manuscript will address many of the reviewer's concerns over robustness and the validation of the methods used to develop the model. For the reviewer's second set of concerns, i.e. whether forces produced by the arrays are responsive to external tensions we direct the reviewer to Fig. 5c where we present more detailed breakdown of the forces produced under different isotropic and anisotropic tensions. In contrast to the strain-hardening response of cross-linked polymer networks, we find the actin networks respond very differently. The pattern of forces that emerge from a remodeled network can focus and amplify due to the unique inclusion of polarized filaments and motors.

This leads to a further question on the dynamics of establishment of this anisotropy. The dynamics of the actomyosin structures is described in the MD simulations, but not *in vivo*. In particular, A-P structures in the mesoderm (Fig 3a) do not seem necessary for anisotropic tension, as they begin to appear toward the end of the laser cut experiment. Would the authors be able to characterise the dynamics either of the reorganisation or of the stress anisotropy? Even if not, I believe the question of the timings must still be mentioned in the main text.

We agree with the reviewer that the dynamics of actomyosin reorganization is important to consider and to correlate with the development of oriented stress across the tissue in our *in silico* model and *in vivo*. We have now added a Supplementary Video (Video 5) that clearly shows how myosin fibers form in a representative *Ctr-RNAi* embryo and how this correlates with the cell shape anisotropy, which is an indicator of stress anisotropy. This shows that the formation of myosin fibers aligned along the a-p axis correlates with an increase in stress anisotropy (that is proportional to the strain) in this direction.

Invagination

The authors highlight that the cup-like invagination of PMG is associated with isotropic tension and actomyosin rings, while furrow-like mesoderm invagination is associated with anisotropic tension and actomyosin longitudinal fiber-like structures. Since they are able to disrupt this latter organisation in RNA assays, it would be of the utmost interest to give some account of the resulting phenotypes in terms of shape of the invagination for these two genetic perturbations.

The overall morphology of ventral furrow in *Spn27A-RNAi* and *Fat2-RNAi* vary from one embryo to another, however, in every cases the furrow is wider and often asynchronous along the a-p axis sometimes resembling a cup-like indentation. One good example of cup-like indentation is shown in Fig S1, the *Spn27A-RNAi* embryo stained for *sim* RNA display two indentations within its extended ventral domain. We have now added a description of furrow shape (p. 14) and compared it with cup-like invagination.

Recommendation

Overall, the manuscript is of extremely high quality both in the research it presents and the way it presents it. I am of the opinion that this manuscript will be a paper of reference in further work on the interplay of global mechanical equilibrium and genetic patterning in morphogenesis, and it is important that the more subtle details of this interplay are fully acknowledged. Thus I recommend a manuscript revision that would discuss these subtler details and possibly include some more quantification of the experimental results presented in this manuscript.

We thank the reviewer for so carefully reading our manuscript and giving us great feedback. We feel that responding to their comments has strengthened our manuscript.

Molecular dynamics simulations.

The appearance of a self-organised ring in the simulations is surprising to me, and was not reported in previous similar models (see references above). Some description of the mechanism would be welcome, as well as a test of its dependence on parameters other than the stiffness (e.g., width of the "boundary region", currently 0.5 μm , but also filament and motor density).

The self-organized ring forms as free filaments and motors encounter filaments that are transiently trapped at the boundary. To our knowledge boundary conditions that represent those found in ventral furrow cells have not been included in previous modeling efforts. Motors binding trapped filaments walk to the plus-ends of these filaments and tend to draw free filaments and other motors to the boundary. Filaments trapped at compliant boundaries can be pulled toward the center of the domain where motors and filament have the potential to contribute to a central aster. With our resubmission we have included a manuscript, Miller et al, that covers the model in greater detail and we hope addresses the reviewers concerns.

Minor comments

Curvature. Pages 5–6, the influence of genetic modifications on the geometry of myosin pattern are discussed, however they also change the curvature of the epithelium, this may be worth mentioning.

We agree with the reviewer and we have added the sentence: *Fat2-RNAi* embryos are shorter along the a-p axis and, because of the altered shape, both ventral and dorsal surfaces have a higher curvature than in wild-type embryos.

F-Actin. Page 8, the authors write "F-actin organization mirrored that of myosin, becoming depleted from the apical center and organized into rings centered within the apical domain (Supplementary Fig. 4a, b)." According to Supp. Fig 4b, in perturbed conditions actin is farther from center than myosin is, and in panel a it is not clear to me that it is structured in rings and that these are away from the cell-cell junctions.

We agree with the reviewer that this might be misleading, we removed from the manuscript 'mirrored that of myosin'. It now read: F-actin organization was depleted from the apical center, such that it resembled a ring with actin enriched around the apical periphery and a hole in the middle of the apical cortex. As shown in the quantifications in Fig. S4, the actin does adopt a ring-like morphology, depleted from the center and enriched in the apical domain around the junctions (and at the junctions as well).

Mechanosensing literature. The authors highlight that the sensing of mechanical conditions in their experiments can be explained without resorting to a transduction step that, however they do not cite the previous literature on this mechanism. In addition to the modeling papers mentioned above, I can suggest experimental work of Mitrossilis et al., PNAS 106:18243, 2009, Trichet et al., PNAS 109:6933, 2012; and modeling work of Zemel et al., J. Phys.: Condens. Matter, 22:194110, 2010, Marcq et al., Biophys. J., 101:L33 2011.

We thank the reviewer for their helpful comments, we added the following citation: Mitrossilis et al., PNAS

106:18243, 2009. However, our model is distinct from these past models in that it predicts the important role that geometry and contraction play, specifically, in orienting molecular motors.

Perijunctional actomyosin. In the end of the discussion, the authors oppose the versatility of medio-apical actin network (studied here) to perijunction actomyosin. However they do not support the claim that perijunctional myosin would not have a similar (although 1D) mechanosensitivity, and indeed Fernandez et al., Dev. Cell, 17:736 have shown some mechanosensitive changes in 1D junctional actomyosin cables.

We agree with the reviewer and acknowledged in the manuscript that actomyosin cables can be mechanosensitive. However, in the example the reviewer refers to, there is an entire genetic pathway that is required to polarize these actomyosin cables in cells (Pare et al., Nature, 2014). Here, we argue that the 2-dimensional and dynamic properties of medial meshworks allow them to adopt different conformations across the apical surface depending on geometric and mechanical context. A cable can't do this.

Laser ablation methods. Page 19, I have difficulty to make out which part of the description applies to straight line and which to circular ablations.

We have added in the manuscript a heading specifying line ablations and circular ablations to describe the two methods.

Molecular dynamics simulations.

In addition to the above comment on ring formation, I believe the simulations would be made more convincing by providing some analyses of the robustness of their results with respect to variations in two classes of parameters : first, the dynamic ones (viscosity, rates of diffusion,...) which are not at all discussed and are (I believe) thought not to have a first order influence, and second on filament and motor numbers.

We refer the reviewer to an included manuscript, Miller et. al., that has just been resubmitted to Molecular Biology of the Cell. The detailed parameter studies suggested by the reviewer are beyond the scope of the current manuscript but comprise a large portion of Miller et. al. From these studies we have developed considerable intuition about the way the actomyosin arrays are formed. Central to that intuition is an understanding of the role of processes that occur on different time-scales. Clearly, fast boundary movements can dominate the resulting shape of actomyosin network. The next most important time-scale is determined by the rate of motor movement on filaments. This rate has two major effects, first, controlling how quickly the filament networks remodel, and second, how quickly the motors are sequestered. The last time-scale is determined by the rate of filament turn-over. For motors and filament movements to reach quasi-equilibrium the filaments must have a long residence time relative to the time it interacts with motors. If filaments turn over quickly then stable patterns of filaments or stable sequestering/depleting of motors cannot occur. A long residence time serves also to transport motors bound with moving/remodeling filaments.

The mechanical coupling of the boundary into the central filament array is also regulated by filament and motor density. To demonstrate this we include two additional simulations (see Rebuttal Figure 3) of conditions where tension is isotropic but where the total number of filaments and motors are either doubled or halved. In the case of double the number of filaments, asters form robustly in the center of the domain. In the case of half the number of filaments and motors, aster formation is irregular and often accompanied by accumulation of filaments at the boundary. As filament density decreases we find the filament/motor network across the domain becomes discontinuous which leads to regions that have been depleted of both filaments and motors.

Rebuttal Figure 3: **effect on filaments and motors density on network conformation**

- a.** Average of 10 simulations showing the averaged network conformation at the final time point. The number of filaments and motors was doubled in these simulations (N=2000 filaments and M=10000 motors)
- b.** Average of 10 simulations showing the averaged network conformation at the final time point. The number of filaments and motors was halved in these simulations (N=500 filaments and M=2500 motors)

The measure of anisotropy in terms of a "maximum" and "minimum force" are unclear to me, is an average force not relevant? If so, why, and would not an increase in filament density allow to use an average?

We thank the reviewer for this question and have worked on several ways to represent the effects of tension anisotropy. It is clear from our simulations that forces emerging from actomyosin arrays under anisotropic tension vary in both magnitude and orientation. Reorganization of arrays under tension focus and amplify the forces produced the arrays. To better illustrate this we now change our reporting of emergent force directionality by plotting the mean ratio of force oriented along the horizontal (stiff) axis over force oriented along the vertical (soft) axis as a function of the resistance asymmetry of the boundary (Figure 5c).

It seems to me not obvious (if not impossible) to create a homogeneous random network of uniform finite length fibers within a finite domain, either the filaments will be aligned with boundaries or their density will be less close to the boundary (the so-called effect of excluded volume). Could the authors document how they initialise their simulation?

We thank the reviewer for mentioning this. We agree with the reviewer that any mechanism that distributes random filaments within a domain and then excludes filaments that cross a wall or barrier would by necessity create an excluded volume beginning 1/2 filament length from the wall. We have revised our description of the method to include this caveat. However, we would point out that the initial organization of the filament array is quickly remodeled.

Reviewer #3 (Remarks to the Author):

This manuscript reports on the impact of tissue geometry on actomyosin organization and on the directionality of cell generated forces. To address this question, the authors use the early *Drosophila* embryo as a model system. During gastrulation, an epithelial furrow forms by contraction of a narrow patch of cells. Using genetic and mechanical perturbations, the author show that the geometry of this patch determines the orientation of actomyosin networks and thus, the anisotropy of mechanical stress at the tissue level.

The question is topical and some of the data are very interesting and convincing. However I consider that the data do not support the main conclusion of the manuscript, which is summarized in the title "actomyosin networks are mechanosensing" enabling "tissue to orient cell forces."

The convincing and very nice part of this manuscript focuses on the effects of a contractile tissue shape onto force generation (Fig 1, 2 and 6). It extends previous observation and suggestions of the same group (Martin et al, 2010) that anisotropy of cell shape changes depends on the geometry of the contractile ventral furrow primordium. This hypothesis was also tested by optogenetic control by Guglielmi et al (2015). Here the authors convincingly show that the geometry of the primordium also determines the anisotropy of stress.

The effect of tissue shape on force generation and actomyosin organization is the main point of our paper. This was not addressed in the past papers that the reviewer mentions. Here, we have changed the geometric and mechanical properties of the tissue while also analyzing the tension and the effects on the cytoskeleton. The Guglielmi paper looked at cell shape, but did not examine tissue forces or cytoskeletal organization; this is what is most interesting and exciting about our findings. The Martin paper analyzed the effects on cell shape by depleting junctions and looking at tissue tears. Therefore, we have defined a new way to examine tissue shape in this system that many will find useful.

Other reviewers agreed about the novelty of our study.

Reviewer #1: The manuscript is very nicely written, and the data are clear and well documented. I found some of the experimental approaches really original (e.g. the RNAi-based changes in mesoderm geometry). The computational model not only helps with the interpretation of the data, but makes valuable predictions, such as the alignment of the myosin motors, that would have been difficult to obtain from experiments.

Reviewer #2: In these papers however, consequences of global tension alteration were shown on tissue dynamics at the scale of a few cells and not in the cytoskeletal arrangement, to the difference of the case of the mesoderm during VF formation studied here.

I don't think that the authors provide evidence that mechanical 'constraints' such as stiffness at the boundary of the contractile tissue (or cells) alter the actomyosin organization and thus force direction. Cells are stretched along the long axis of the contractile tissue and it is likely that the actomyosin network reorganizes accordingly. The fig. 3a (and 3c) mainly show that the actomyosin network organization anisotropy follows the anisotropy in cell shape. In Fig 3a, the actomyosin network can be seen as "fibers" in squeezed cells but also as rings in more isotropic cells.

As a consequence, while interesting, a cellular model assuming that the cell boundary has an anisotropic resistance (or stiffness) has little relevance to the present study.

We disagree with a couple point here. First, cells are not stretched along the long axis. They simply fail to constrict as much along that axis, because there is higher tension. Second, what is being sensed is the failure of the network to contract as much along that particular axis, which is impacted by the resistance of the surrounding tissue. We thank the reviewer for pointing this out, and we have revised the text to make this point more clear. Third, the reviewer is incorrect in thinking that isotropic wild-type cells exhibit rings that resemble those formed in *Spn27A-RNAi* or the PMG. We have described the network in great detail and myosin forms a supracellular meshwork (Martin et al., 2010). Because myosin fibers connect between cells, there can be loops involving multiple cells, this is possibly what the reviewer sees. We have examined thousands of wild-type cells very carefully and the organization of meshworks in these and *Spn27A-RNAi* cells are distinct. Finally we have now tested the effect of confinement in our *in silico* model on force production and found that it doesn't explain what we observed *in vivo*, indeed, simulated networks embedded within ellipsoidal shape directed force along the short axis (see Rebuttal Figure 1).

Anisotropic spatial confinement has been shown to organize actin networks *in vitro* (Marina Soares e Silva, *Soft Matter*, 2011). Alternative explanations are likely to be more justified than the one proposed by the authors : the observed actomyosin organization could be induced by the stretching of the cells along the a-p axis in the contractile tissue. The authors only allude to this possibility in their discussion (and references 2 & 3). I would expect the authors explore this mechanism thoroughly. If not, they have to test their model through a series of new experiments, in which stiffness at tissue boundaries would be modified (independent of cell shape).

We disagree that the Soares e Silva model is relevant to what we have observed or to our model. In this system, exceedingly long, stable actin filaments ($> 6 \mu\text{m}$ length) are placed in confinement and are shown to bundle. This is not relevant to our system for several reasons: 1) There is no myosin in these experiments, thus, the Soares e Silva system is not a model of contraction; 2) The actin filaments are stabilized with phalloidin to make them stiffer. Phalloidin is not normally decorating actin filaments in cells. 3) The actin filaments in this experiment are much longer than what would be physiologically relevant. Actin filaments and cables observed in ventral furrow cells are shorter than the cell diameter (Mason et al., 2013) and, in general, the length of actin filaments *in vivo* are thought to be $< 1 \mu\text{m}$ much shorter than the diameter of even a constricted cell (Cano et al., JCB, 1991; Podolski and Steck, JBC, 1990). 4) Actin filaments undergo turnover *in vivo*, which has been shown to be the case for ventral furrow cells (Jodoin et al., *Dev. Cell*, 2015). There is no turnover in this biochemical reconstitution, which indeed would make them confined.

The manuscript concludes that the actomyosin meshworks adapt to external mechanical pattern and redirect cell forces. However the stress anisotropy comes from the very cells which are contractile in the furrow primordium. I don't see the external mechanical pattern and at this stage, the conclusions drawn by the authors are not supported by experimental evidence.

The geometry of the contractile tissue generates a mechanical pattern that cells within this contractile domain can feel. We have demonstrated that this effect is related to the shape of the embryo, because round embryos do not exhibit the same pattern. In an ellipsoidal embryo that has viscoelastic properties it would probably be difficult to pull cells around a region of high curvature (i.e. the poles of the embryo). We have demonstrated that cells “feel” and respond to this mechanical pattern in multiple ways: 1) Changing the mechanical context of the tissue with either *Spn27A-RNAi* or *Fat2-RNAi* results in myosin rings, similar to the PMG. In addition, *Spn27A-RNAi* embryos often exhibit cup-like indentations, like those in Figure S1 – *sim* in situ. We have added this point to the text (p.14). 2) We have performed a mechanical suppression of the *Spn27A-RNAi* phenotype, where these line ablation experiments are most cleanly done (Figure 4a-d). 3) We can promote myosin ring formation by making vertical line ablations in a wild-type embryo (Figure 4f-i).

REVIEWERS' COMMENTS:

Reviewer #1 (Remarks to the Author):

The authors have addressed most of my concerns. However, they have chosen not to include in the manuscript a number of their answers, which I think could confuse or mislead the readers. Thus, I would ask that the published version of the work includes, even as supplementary material, the following items:

1. The percentage of lethality caused by the different RNAi manipulations (see response to Major Point #3). This is important for the readers to evaluate the severity of the phenotypes.
2. The simulation results testing the effect of confinement (see rebuttal figure 1). I noticed Reviewer 3 referenced (Soares e Silva, 2011). A number of readers are also likely to wonder about a role for confinement in the organization of the cytoskeleton during mesoderm invagination. Without rebuttal figure 1, readers may question the model put forward by the authors (and the job of this reviewer!).
3. The effects of expressing a motor-deficient myosin (see rebuttal figure 2). Showing that loss of myosin activity affects the organization of ROCK at least partially addresses which of the molecules are more responsive to mechanical factors. While the insight provided by the model is valuable, wet lab experiments are invaluable.

Also, I understand the arguments provided by the authors to only qualitatively compare pulsing in the endoderm and in the mesoderm. The authors should modify the first sentence of page 14 to be: "Myosin contraction exhibited pulses qualitatively similar to myosin in the VF (Supplementary Fig. 7)."

Reviewer #2 (Remarks to the Author):

This revised manuscript clearly demonstrates how, complementary to the biochemical activity pattern governed by genetic expression, the global mechanical balance (and thus the geometry) of tissue mediates phenotypical changes in cells, namely cell morphology and actomyosin cytoskeleton arrangement. Experiments where either the geometry of the tissues or the patterning of activity differ in a controlled manner show that these tissue-scale changes feed back on cell phenotype far away from the areas that have been perturbed. The original phenotype is rescued when mechanical balance is restored using laser cuts or tissue tearing, showing that these phenotypical changes are indeed caused by alterations of the tension felt by cells. Molecular dynamics simulations propose a plausible mechanism to explain phenotypical changes of the actin and myosin within cells subjected to different mechanical conditions. The new Supp. Video 5 is convincing, and supports the idea that indeed as cells deform their cytoskeleton is also being reorganised and aligned with a specific direction, which is the one along which tension is maximal and convergent rate of strain minimal.

Since actomyosin is known to be causal of the appearance of tension and/or morphogenetic movements in the tissues studied here (*Drosophila* mesoderm and post-mid gut endoderm), the authors link the observed tension or movement with the actomyosin organisation. This was the weaker and more debatable conclusion in the initial manuscript. The revision includes new experimental data and some more details on molecular dynamics simulation makes this more convincing. Specifically, the authors do show that global geometry and mechanical balance lead to an anisotropic stress within cells. Using MD simulations where the anisotropy is implemented through anisotropic stiffness of boundary conditions, they also show that the anisotropic stress is likely to be due to actomyosin mechanosensing. This is consistent with what one might expect

from the literature on mechanosensing in isotropic conditions. Experimentally, laser ablations give further evidence of the anisotropy of stress. This justifies the research result as stated in the title of the manuscript.

However, the fact that the anisotropy of the stress locally (or 'cell force directionality') would be a consequence of the reorganisation of the cytoskeleton is not clearly proven, either experimentally or in simulations. The authors deduce this from the fact that, in MD simulations, the filaments reorganise in an anisotropic manner and simultaneously exert an anisotropic stress. However, this does not mean that this anisotropic stress crucially depends on the anisotropic reorganisation. Simulations should allow the authors to answer this question, e.g. by plotting a figure equivalent to Fig 5f for the first instants of the simulation, before the network reorganises. If the anisotropy of forces appears ahead of the reorganisation, then it would mean that the boundary conditions govern it directly rather than the meshwork reorganisation. I am of the opinion that the core results of the research would not be modified by this. However, in the absence of a formal proof of it, the flowchart in Fig 7 should be modified as it is not proven that it is through 'Meshwork reorganisation' that the 'Mechanical pattern' influences the 'Force directionality'.

Summary and recommendation: The research presented in the manuscript constitutes very significant advance in the fine understanding of how geometry and mechanics modify the actomyosin meshwork organisation on the one hand, and give rise to anisotropic stress on the other hand. The causality between these two effects should be presented as a discussion hypothesis or specifically supported. With this change done, I recommend this manuscript for publication.

Reviewer #3 (Remarks to the Author):

The authors have addressed most of the points I raised. They provide better evidence and explanation of how the shape and geometry of a contractile tissue impact on cell shape changes and the orientation of contractile networks within cells.

I still find a disconnect between the experiments and the simulations. The simulations address the question of how anisotropic mechanical boundary conditions (anisotropic compliance) influence the organization/orientation of actomyosin networks, while the experiments focus on the role of tension anisotropy and geometry of the contractile tissue. Please clarify how to connect the two.

Reviewer #1 (Remarks to the Author):

The authors have addressed most of my concerns. However, they have chosen not to include in the manuscript a number of their answers, which I think could confuse or mislead the readers. Thus, I would ask that the published version of the work includes, even as supplementary material, the following items:

1. The percentage of lethality caused by the different RNAi manipulations (see response to Major Point #3). This is important for the readers to evaluate the severity of the phenotypes.

This information is included in the main text, see p.6 : “Interestingly, the majority of Fat2-RNAi embryos hatched ($n = 88/100$ Fat2-RNAi embryos, compared to $n = 97/100$ wild-type embryos), giving rise to larvae that were phenotypically similar, but smaller than wild-type larvae. This suggests that the round shape of the embryo did not prevent further embryogenesis. *Spn27A* loss of function however is lethal (Ligoxygakis et al. 2003).”

2. The simulation results testing the effect of confinement (see rebuttal figure 1). I noticed Reviewer 3 referenced (Soares e Silva, 2011). A number of readers are also likely to wonder about a role for confinement in the organization of the cytoskeleton during mesoderm invagination. Without rebuttal figure 1, readers may question the model put forward by the authors (and the job of this reviewer!).

Because two reviewers have now pointed this out, we have now included this result in Supplementary Figure 6c. The Soares e Silva paper (2011) is of limited relevance to our *in vivo* system because: 1) The length scales of the filaments is much longer $>6 \mu\text{m}$ than is physiologically relevant, 2) The filaments are stabilized with phalloidin, which makes their bending stiffness greater, and most importantly, 3) There is actin filament turnover *in vivo* and in our MD simulation. Even so, we have now included a short paragraph in the Results section discussing the possible effect of cell shape and confinement (see p.13).

3. The effects of expressing a motor-deficient myosin (see rebuttal figure 2). Showing that loss of myosin activity affects the organization of ROCK at least partially addresses which of the molecules are more responsive to mechanical factors. While the insight provided by the model is valuable, wet lab experiments are invaluable.

We agree that this is an interesting result, but one in which we have found to be confusing and circular to others we have asked to comment on the manuscript. We have considered whether or not to include this result at both submission and resubmission and decided against doing it. We do not fully agree with the interpretation that this result shows that ROCK is “more responsive to mechanical factors”. This is because, ROCK organization and localization are not affected by itself in the *sqh-TA* mutant, but rather the entire actomyosin organization (which is what is the most affected in the myosin mutant). The difficulty in interpreting this result comes from the fact that we cannot separate the role of myosin in force production vs. in force sensing. The manuscript already shows that myosin and ROCK change together in response to different geometric constraints. In addition, both our lab and others have shown that apical ROCK localization is dependent on actin and myosin (Munjaj et al., 2015; Coravos and Martin, 2016). Therefore, we don't think adding this result to the main text adds to the manuscript. We have opted for our rebuttal to be made public upon publication for the aficionados in the field to see.

Also, I understand the arguments provided by the authors to only qualitatively compare pulsing in the endoderm and in the mesoderm. The authors should modify the first sentence of page 14 to be: “Myosin contraction exhibited pulses qualitatively similar to myosin in the VF (Supplementary Fig. 7).”

We have made this modification. Thank you for your excellent comments and for helping us improve our manuscript.

Reviewer #2 (Remarks to the Author):

This revised manuscript clearly demonstrates how, complementary to the biochemical activity pattern governed by genetic expression, the global mechanical balance (and thus the geometry) of tissue mediates phenotypical changes in cells, namely cell morphology and actomyosin cytoskeleton arrangement. Experiments where either the geometry of the tissues or the patterning of activity differ in a controlled manner show that these tissue-scale changes feed back on cell phenotype far away from the areas that have been perturbed. The original phenotype is rescued when mechanical balance is restored using laser cuts or tissue tearing, showing that these phenotypical changes are indeed caused by alterations of the tension felt by cells. Molecular dynamics simulations propose a plausible mechanism to explain phenotypical changes of the actin and myosin within cells subjected to different mechanical conditions. The new Supp. Video 5 is convincing, and supports the idea that indeed as cells deform their cytoskeleton is also being reorganised and aligned with a specific direction, which is the one along which tension is maximal and convergent rate of strain minimal.

Since actomyosin is known to be causal of the appearance of tension and/or morphogenetic movements in the tissues studied here (Drosophila mesoderm and post-mid gut endoderm), the authors link the observed tension or movement with the actomyosin organisation. This was the weaker and more debatable conclusion in the initial manuscript. The revision includes new experimental data and some more details on molecular dynamics simulation makes this more convincing. Specifically, the authors do show that global geometry and mechanical balance lead to an anisotropic stress within cells. Using MD simulations where the anisotropy is implemented through anisotropic stiffness of boundary conditions, they also show that the anisotropic stress is likely to be due to actomyosin mechanosensing. This is consistent with what one might expect from the literature on mechanosensing in isotropic conditions. Experimentally, laser ablations give further evidence of the anisotropy of stress. This justifies the research result as stated in the title of the manuscript.

However, the fact that the anisotropy of the stress locally (or 'cell force directionality') would be a consequence of the reorganisation of the cytoskeleton is not clearly proven, either experimentally or in simulations. The authors deduce this from the fact that, in MD simulations, the filaments reorganise in an anisotropic manner and simultaneously exert an anisotropic stress. However, this does not mean that this anisotropic stress crucially depends on the anisotropic reorganisation. Simulations should allow the authors to answer this question, e.g. by plotting a figure equivalent to Fig 5f for the first instants of the simulation, before the network reorganises. If the anisotropy of forces appears ahead of the reorganisation, then it would mean that the boundary conditions govern it directly rather than the meshwork reorganisation. I am of the opinion that the core results of the research would not be modified by this. However, in the absence of a formal proof of it, the flowchart in Fig 7 should be modified as it is not proven that it is through 'Meshwork reorganisation' that the 'Mechanical pattern' influences the 'Force directionality'.

We understand the reviewer to be concerned about whether the reorganization of the cytoskeleton in and of itself is the cause of the 'force directionality' in the tissue. We are hesitant to draw strong conclusions from the earliest time points of the simulation because motors and filaments are randomly dispersed with random orientations. At that time, the network is mostly contiguous and geometrically isotropic. To carry out the simulation suggested by the reviewer would require us to investigate dynamic transitions between quasi-stable networks as boundary tensions or cell shape is changed. We agree that force directionality and thus, mechanical constraints, impact the organization of the actomyosin meshwork, which argues for a mechanosensing function we proposed. We speculate that this mechanosensing function reinforces the initial mechanical pattern, causing cells to direct force along a given axis. We have slightly modified the flowchart to communicate that actomyosin reorganization is the result of the mechanical constraints and that we speculate that this mechanosensing function could reinforce directional force generation. In addition, we have modified the Discussion to better explain this model. For instance, we added the following sentence, "Because actomyosin meshwork organization cannot be altered independent of tissue mechanics (and constraints) it is impossible to assess the direct contribution of the reorganization, but our *in silico* simulations predict how this might occur (see below)."

Summary and recommendation: The research presented in the manuscript constitutes very significant advance in the fine understanding of how geometry and mechanics modify the actomyosin meshwork organisation on the one hand, and give rise to anisotropic stress on the other hand. The causality between these two effects should

be presented as a discussion hypothesis or specifically supported. With this change done, I recommend this manuscript for publication.

We thank the reviewer for critically reading our manuscript and for their helpful feedback and suggestions.

Reviewer #3 (Remarks to the Author):

The authors have addressed most of the points I raised. They provide better evidence and explanation of how the shape and geometry of a contractile tissue impact on cell shape changes and the orientation of contractile networks within cells.

I still find a disconnect between the experiments and the simulations. The simulations address the question of how anisotropic mechanical boundary conditions (anisotropic compliance) influence the organization/orientation of actomyosin networks, while the experiments focus on the role of tension anisotropy and geometry of the contractile tissue. Please clarify how to connect the two.

Please see our response to Reviewer #2. In short, our experiments show that actomyosin organization senses and responds to anisotropic mechanical constraints. In a tissue, we showed this results from the shape of the contractile domain. This, in turn, orients force generation, which we speculate could reinforce force directionality in the tissue. We have added text in the Discussion to better connect our results from the simulations and our *in vivo* results.

p.17: "This prediction can be translated to the tissue level, where cells embedded in a tissue also can experience anisotropic constraints (e.g., more cells constricting along the a-p axis). In the ventral furrow, the greater resistance to constriction along the a-p axis can lead to polarization and alignment of the actomyosin cortex. In turn, aligned motors and directed cell force, would further promote anisotropic epithelial tension."

In addition, we added a new paragraph to the Discussion (2nd paragraph), which specifically outlines how we envision the ventral furrow forming. We think this should clarify the relationship between the model and the *in vivo* system.

Again, we thank the reviewer for their helpful comments and the time they have taken with our manuscript.